# EpiGRAF: Rethinking training of 3D GANs

**Ivan Skorokhodov**
KAUST

**Sergey Tulyakov**
Snap Inc.

**Yiqun Wang**
KAUST

**Peter Wonka**
KAUST

## Abstract

A recent trend in generative modeling is building 3D-aware generators from 2D image collections. To induce the 3D bias, such models typically rely on volumetric rendering, which is expensive to employ at high resolutions. Over the past months, more than ten works have addressed this scaling issue by training a separate 2D decoder to upsample a low-resolution image (or a feature tensor) produced from a pure 3D generator. But this solution comes at a cost: not only does it break multi-view consistency (i.e., shape and texture change when the camera moves), but it also learns geometry in low fidelity. In this work, we show that obtaining a high-resolution 3D generator with SotA image quality is possible by following a completely different route of simply training the model patch-wise. We revisit and improve this optimization scheme in two ways. First, we design a location- and scale-aware discriminator to work on patches of different proportions and spatial positions. Second, we modify the patch sampling strategy based on an annealed beta distribution to stabilize training and accelerate the convergence. The resulting model, named EpiGRAF, is an efficient, high-resolution, pure 3D generator, and we test it on four datasets (two introduced in this work) at $256^2$ and $512^2$ resolutions. It obtains state-of-the-art image quality, high-fidelity geometry and trains $\approx 2.5\times$ *faster* than the upsampler-based counterparts.

Code/data/visualizations: https://universome.github.io/epigraf

## 1 Introduction

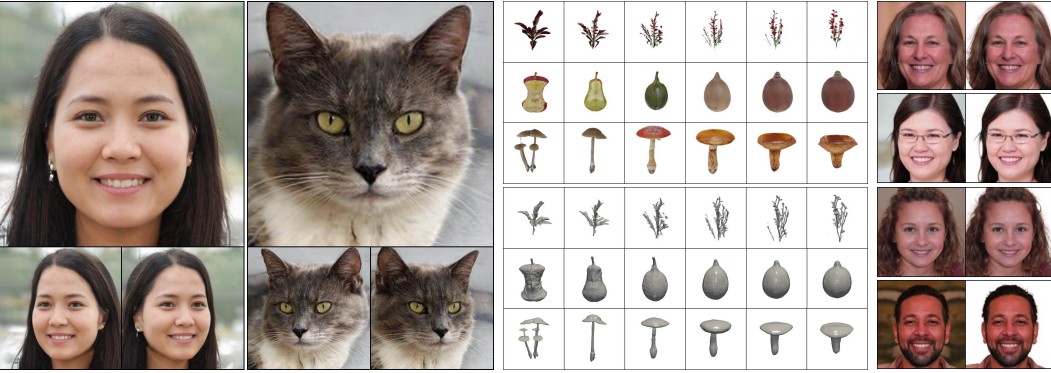

Figure 1: We build a pure NeRF-based generator trained in a patch-wise fashion. Left two grids: samples on FFHQ $512^2$ [25] and Cats $256^2$ [77]. Middle grids: interpolations between samples on M-Plants and M-Food (upper) and corresponding geometry interpolations (lower). Right grid: background separation examples. In contrast to the upsampler-based methods, one can naturally incorporate the techniques from the traditional NeRF literature into our generator: for background separation, we simply copy-pasted the corresponding code from NeRF++ [76].

36th Conference on Neural Information Processing Systems (NeurIPS 2022).

Generative models for image synthesis achieved remarkable success in recent years and now enjoy a lot of practical applications [55, 24]. While initially they mainly focused on 2D images [21, 66, 25, 4, 28], recent research explored generative frameworks with partial 3D control over the underlying object in terms of texture/structure decomposition, novel view synthesis or lighting manipulation (e.g., [58, 56, 7, 68, 6, 12, 49]). These techniques are typically built on top of the recently emerged neural radiance fields (NeRF) [38] to explicitly represent the object (or its latent features) in 3D space.

NeRF is a powerful framework, which made it possible to build expressive 3D-aware generators from challenging RGB datasets [7, 12, 6]. Under the hood, it trains a multi-layer perceptron (MLP) $\mathsf{F}(\boldsymbol{x}; \boldsymbol{d}) = (\boldsymbol{c}, \sigma)$ to represent a scene by encoding a density $\sigma \in \mathbb{R}_+$ for each coordinate position $\boldsymbol{x} \in \mathbb{R}^3$ and a color value $\boldsymbol{c} \in \mathbb{R}^3$ from $\boldsymbol{x}$ and view direction $\boldsymbol{d} \in \mathbb{S}^2$ [38]. To synthesize an image, one renders each pixel independently by casting a ray $\boldsymbol{r}(q) = \boldsymbol{o} + q\boldsymbol{d}$ (for $q \in \mathbb{R}_+$) from origin $\boldsymbol{o} \in \mathbb{R}^3$ into the direction $\boldsymbol{d} \in \mathbb{S}^2$ and aggregating many color values along it with their corresponding densities. Such a representation is very expressive but comes at a cost: rendering a single pixel is computationally expensive and makes it intractable to produce a lot of pixels in one forward pass. It is *not* fatal for reconstruction tasks where the loss can be robustly computed on a subset of pixels, but it creates significant scaling problems for generative NeRFs: they are typically formulated in a GAN-based framework [14] with 2D convolutional discriminators requiring an entire image as input.

People address these scaling issues of NeRF-based GANs in different ways. The dominating approach is to train a separate 2D decoder to produce a high-resolution image from a low-resolution image or feature grid rendered from a NeRF backbone [43]. During the past six months, there appeared *more than a dozen* of methods that follow this paradigm (e.g., [6, 15, 71, 47, 79, 35, 75, 23, 72, 78, 64]). While using the upsampler allows scaling the model to high resolution, it comes with two severe limitations: 1) it breaks the multi-view consistency of a generated object, i.e., its texture and shape change when the camera moves; and 2) the geometry gets only represented in a low resolution ($\approx 64^3$). In our work, we show that by dropping the upsampler and using a simple patch-wise optimization scheme, one can build a 3D generator with better image quality, faster training speed, and without the above limitations.

Patch-wise training of NeRF-based GANs was initially proposed by GRAF [56] and got largely neglected by the community since then. The idea is simple: instead of training the generative model on full-size images, one does this on small random crops. Since the model is coordinate-based [59, 65], it does not face any issues to synthesize only a subset of pixels. This serves as an excellent way to save computation for both the generator and the discriminator since it makes them both operate on patches of small spatial resolution. To make the generator learn both the texture and the structure, crops are sampled to be of variable scales (but having the *same* number of pixels). In some sense, this can be seen as optimizing the model on low-resolution images + high-resolution patches.

In our work, we improve patch-wise training in two crucial ways. First, we redesign the discriminator by making it better suited to operating on image patches of variable scales and locations. Convolutional filters of a neural network learn to capture different patterns in their inputs depending on their semantic receptive fields [30, 46]. That's why it is detrimental to reuse the same discriminator to judge both high-resolution local and low-resolution global patches, inducing additional burden on it to mix filters' responses of different scales. To mitigate this, we propose to modulate the discriminator's filters with a hypernetwork [16], which predicts which filters to suppress or reinforce from a given patch scale and location.

Second, we change the random scale sampling strategy from an annealed uniform to an annealed beta distribution. Typically, patch scales are sampled from a uniform distribution $s \sim \mathcal{U}[s(t), 1]$ [56, 36, 5], where the minimum scale $s(t)$ is gradually decreased (i.e. annealed) till some iteration $T$ from $s(0) = 0.9$ to a smaller value $s(T)$ (in the interval $[0.125 - 0.5]$) during training. This sampling strategy prevents learning high-frequency details early on in training and puts too little attention on the structure after $s(t)$ reaches its final value $s(T)$. This makes the overall convergence of the generator slower and less stable that's why we propose to sample patch scales using the beta distribution $\text{Beta}(1, \beta(t))$ instead, where $\beta(t)$ is gradually annealed from $\beta(0) \approx 0$ to some maximum value $\beta(T)$. In this way, the model starts learning high-frequency details immediately with the start of training and focuses more on the structure after the growth finishes. This simple change stabilizes the training and allows it to converge faster than the typically used uniform distribution [56, 5, 36].

EG3D                                    Ours

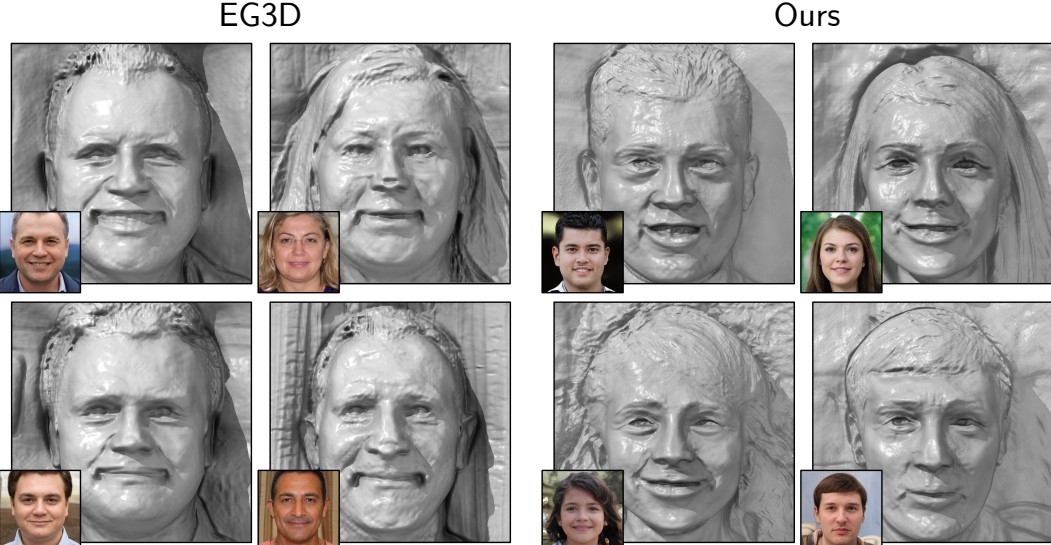

Figure 2: Comparing the geometry between EG3D [6] and our generator on FFHQ $512^2$. For each method, we computed the density field in the $512^3$ volume resolution and extracted the surfaces using marching cubes. The geometry of our generator contains more high-frequency details (e.g., hair strands are better separated) since it learns it in full resolution. EG3D uses the $64^2$ rendering resolution (and $128^2$ during the last 10% of the training) so its shapes appear over-smoothed.

We use those two ideas to develop a novel state-of-the-art 3D GAN: Efficient patch-informed Generative Radiance Fields (EpiGRAF). We employ it for high-resolution 3D-aware image synthesis on four datasets: FFHQ [25], Cats [77], Megascans Plants, and Megascans Food. The last two benchmarks are introduced in our work and contain $360°$ renderings of photo-realistic scans of different plants and food objects (described in §4). They are much more complex in terms of geometry and are well-suited for assessing the structural limitations of modern 3D-aware generators.

Our model uses a pure NeRF-based backbone, that's why it represents geometry in high resolution and does not suffer from multi-view synthesis artifacts, as opposed to upsampler-based generators. Moreover, it has higher or comparable image quality (as measured by FID [20]) and $2.5\times$ lower training cost. Also, in contrast to upsampler-based 3D GANs, our generator can naturally incorporate the techniques from the traditional NeRF literature. To demonstrate this, we incorporate background separation into our framework by simply copy-pasting the corresponding code from NeRF++ [76].

## 2   Related work

**Neural Radiance Fields**. Neural Radiance Fields (NeRF) is an emerging area [38], which combines neural networks with volumetric rendering techniques to perform novel-view synthesis [38, 76, 2], image-to-scene generation [74], surface reconstruction [45, 69, 44] and other tasks [9, 17, 50]. In our work, we employ them in the context of 3D-aware generation from a dataset of RGB images [56, 7].

**3D generative models**. A popular way to learn a 3D generative model is to train it on 3D data or in an autoencoder's latent space (e.g., [10, 70, 1, 34, 31, 39, 29]). This requires explicit 3D supervision and there appeared methods which train from RGB datasets with segmentation masks, keypoints or multiple object views [13, 32, 54]. Recently, there appeared works which train from single-view RGB only, including mesh-generation methods [19, 73, 53] and methods that extract 3D structure from pretrained 2D GANs [58, 48]. And recent neural rendering advancements allowed to train NeRF-based generators [56, 7, 42] from purely RGB data from scratch, which became the dominating direction since then and which are typically formulated in the GAN-based framework [14].

**NeRF-based GANs**. HoloGAN [41] generates a 3D feature voxel grid which is projected on a plane and then upsampled. GRAF [56] trains a noise-conditioned NeRF in an adversarial manner. $\pi$-GAN [7] builds upon it and uses progressive growing and hypernetwork-based [16] conditioning

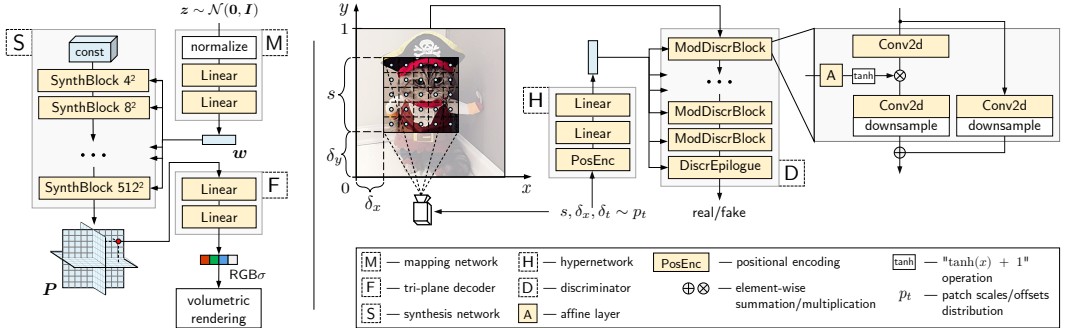

Figure 3: Our generator (left) is purely NeRF-based and uses the tri-plane backbone [6] with the StyleGAN2 [26] decoder (but without the 2D upsampler). Our discriminator (right) is also based on StyleGAN2, but is modulated by the patch location and scale parameters. We use the patch-wise optimization for training [56] with our proposed Beta scale sampling, which allows our model to converge ×2-3 faster than the upsampler-based architectures despite the generator modeling geometry in full resolution (see Tab 1).

in the generator. GRAM [12] builds on top of $\pi$-GAN and samples ray points on a set of learnable iso-surfaces. GNeRF [36] adapts GRAF for learning a scene representation from RGB images without known camera parameters. GIRAFFE [43] uses a composite scene representation for better controllability. CAMPARI [42] learns a camera distribution and a background separation network with inverse sphere parametrization [76]. To mitigate the scaling issue of volumetric rendering, many recent works train a 2D decoder under different multi-view consistency regularizations to upsample a low-resolution volumetrically rendered feature grid [6, 15, 71, 47, 79, 72, 78]. However, none of such regularizations can currently provide the multi-view consistency of pure-NeRF-based generators.

**Patch-wise generative models**. Patch-wise training had been routinely utilized to learn the textural component of image distribution when the global structure is provided from segmentation masks, sketches, latents or other sources (e.g., [22, 57, 11, 67, 52, 51, 33, 61]). Recently, there appeared works which sample patches at variable scales, in which way a patch can carry global information about the whole image. Recent works use it to train a generative NeRF [56], fit a neural representation in an adversarial manner [36] or to train a 2D GAN on a dataset of variable resolution [5].

## 3   Model

We build upon StyleGAN2 [26], replacing its generator with the tri-plane-based NeRF model [6] and using its discriminator as the backbone. We train the model on $r \times r$ patches (we use $r = 64$ everywhere) of random scales instead of the full images of resolution $R \times R$. Scales $s \in [\frac{r}{R}, 1]$ are randomly sampled from a time-varying distribution $s \sim p_t(s)$.

### 3.1   3D generator

Compared to upsampler-based 3D GANs [15, 43, 72, 79, 6, 78], we use a pure NeRF [38] as our generator G and utilize the tri-plane representation [6, 8] as the backbone. It consists of three components: 1) mapping network M : $z \mapsto w$ which transforms a noise vector $z \sim \mathbb{R}^{512}$ into the latent vector $w \sim \mathbb{R}^{512}$; 2) synthesis network S : $w \mapsto P$ which takes the latent vector $w$ and synthesizes three 32-dimensional feature planes $P = (P_{xy}, P_{yz}, P_{xz})$ of resolution $R_p \times R_p$ (i.e. $P_{(*)} \in \mathbb{R}^{R_p \times R_p \times 32}$); 3) tri-plane decoder network F : $(x, P) \mapsto (c, \sigma) \in \mathbb{R}^4$, which takes the space coordinate $x \in \mathbb{R}^3$ and tri-planes $P$ as input and produces the RGB color $c \in \mathbb{R}^3$ and density value $\sigma \in \mathbb{R}_+$ at that point by interpolating the tri-plane features in the given coordinate and processing them with a tiny MLP. In contrast to classical NeRF [38], we do not utilize view direction conditioning since it worsens multi-view consistency [7] in GANs, which are trained on RGB datasets with a single view per instance. To render a single pixel, we follow the classical volumetric rendering pipeline with hierarchical sampling [38, 7], using 48 ray steps in coarse and 48 in fine sampling stages. See the accompanying source code for more details.

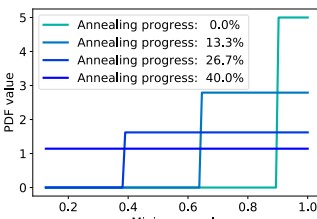 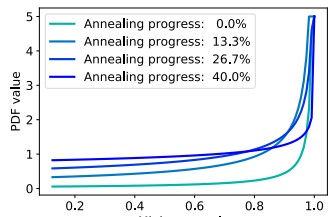 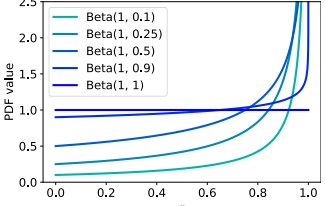

Figure 4: Comparing uniform (left) and beta (middle) annealed patch scale sampling in terms of their probability density function (PDF) (for visualization purposes, we clamp the maximum density value to 5); (right) PDF of $\text{Beta}(1, \beta)$, provided for completeness. Uniform distribution with annealed $s_{\min}(0) = 0.9$ from 0.9 to $s_{\min}(T) = 0.125$ does not put any attention to high-frequency details in the beginning and treats small-scale and large-scale patches equally at the end of the annealing. Beta distribution with annealed $\beta(0) \approx 0$ to $\beta(T) \approx 1$, in contrast, lets the model learn high-resolution texture immediately after the training starts, and puts more focus on the structure at the end.

## 3.2 2D scale/location-aware discriminator

Our discriminator D is built on top of StyleGAN2 [26]. Since we train the model in a patch-wise fashion, the original backbone is not well suited for this: convolutional filters are forced to adapt to signals of very different scales and extracted from different locations. A natural way to resolve this problem is to use separate discriminators depending on the scale, but that strategy has three limitations: 1) each particular discriminator receives less overall training signal (since the batch size is limited); 2) from an engineering perspective, it is more expensive to evaluate a convolutional kernel with different parameters on different inputs; 3) one can use only a small fixed amount of possible patch scales. This is why we develop a novel hypernetwork-modulated [16, 62] discriminator architecture to operate on patches with continuously varying scales.

To modulate the convolutional kernels of D, we define a hypernetwork $\text{H} : (s, \delta_x, \delta_y) \mapsto (\boldsymbol{\sigma}_1, ..., \boldsymbol{\sigma}_L)$ as a 2-layer MLP with $\mathsf{tanh}$ non-linearity at the end which takes patch scale $s$ and its cropping offsets $\delta_x, \delta_y$ as input and produces modulations $\boldsymbol{\sigma}_\ell \in (0, 2)^{c_{\text{out}}^\ell}$ (we shift the $\mathsf{tanh}$ output by 1 to map into the 1-centered interval), where $c_{\text{out}}^\ell$ is the number of output channels in the $\ell$-th convolutional layer. Given a convolutional kernel $\boldsymbol{W}^\ell \in \mathbb{R}^{c_{\text{out}}^\ell \times c_{\text{in}}^\ell \times k \times k}$ and input $\boldsymbol{x} \in \mathbb{R}^{c_{\text{in}}}$, a straightforward strategy to apply the modulation is to multiply $\boldsymbol{\sigma}$ on the weights (depicting the convolution operation by $\mathsf{conv2d}(.)$ and omitting its other parameters for simplicity):

$$\boldsymbol{y} = \mathsf{conv2d}(\boldsymbol{W}^\ell \odot \boldsymbol{\sigma}, \boldsymbol{x}), \tag{1}$$

where we broadcast the remaining axes and $\boldsymbol{y} \in \mathbb{R}^{c_{\text{out}}}$ is the layer output (before the non-linearity). However, using different kernel weights on top of different inputs is inefficient in modern deep learning frameworks (even with the group-wise convolution trick [26]). That's why we use an equivalent strategy of multiplying the weights on $\boldsymbol{x}$ instead:

$$\boldsymbol{y} = \boldsymbol{\sigma} \odot \mathsf{conv2d}(\boldsymbol{W}^\ell, \boldsymbol{x}). \tag{2}$$

This suppresses and reinforces different convolutional filters of the layer depending on the patch scale and location. And to incorporate even stronger conditioning, we also use the projection strategy [40] in the final discriminator block. We depict our discriminator architecture in Fig 3. As we show in Tab 2, it allows us to obtain $\approx 15\%$ lower FID compared to the standard discriminator.

## 3.3 Patch-wise optimization with Beta-distributed scales

Training NeRF-based GANs is computationally expensive because rendering each pixel via volumetric rendering requires many evaluations (e.g., in our case, 96) of the underlying MLP. For scene reconstruction tasks, it does not create issues since the typically used $\mathcal{L}_2$ loss [38, 76, 69] can be robustly computed on a sparse subset of the pixels. But for NeRF-based GANs, it becomes prohibitively expensive for high resolutions since convolutional discriminators operate on dense full-size images. The currently dominating approach to mitigate this is to train a separate 2D decoder to upsample a low-resolution image representation rendered from a NeRF-based MLP. But this breaks multi-view consistency (i.e., object's shape and texture change when the camera is moving) and

learns the 3D geometry in a low resolution (from $\approx 16^2$ [72] to $\approx 128^2$ [6]). This is why we build upon the multi-scale patch-wise training scheme [56] and demonstrate that it can give state-of-the-art image quality and training speed without the above limitations.

Patch-wise optimization works the following way. On each iteration, instead of passing the full-size $R \times R$ image to D, we instead input only a small patch with resolution $r \times r$ of random scale $s \in [r/R, 1]$ and extracted with a random offset $(\delta_x, \delta_y) \in [0, 1 - s]^2$. We illustrate this procedure in Fig 3. Patch parameters are sampled from distribution:

$$s, \delta_x, \delta_y \sim p_t(s, \delta_x, \delta_y) \triangleq p_t(s)p(\delta_x|s)p(\delta_y|s) \qquad (3)$$

where $t$ is the current training iteration. In this way, patch scales depend on the current training iteration $t$, and offsets are sampled independently after we know $s$. As we show next, the choice of distribution $p_t(s)$ has a crucial influence on the learning speed and stability.

Typically, patch scales are sampled from the annealed uniform distribution [56, 36, 5] $s$:

$$p_t(s) = U[s_{\min}(t), 1], \qquad s_{\min}(t) = \texttt{lerp}\left[1, r/R, \min(t/T, 1)\right], \qquad (4)$$

where $\texttt{lerp}$ is the linear interpolation function[1], and the left interval bound $s_{\min}(t)$ is gradually annealed during the first $T$ iterations until it reaches the minimum possible value of $r/R$.[2] But this strategy does not let the model learn high-frequency details early on in training and puts little focus on the structure when $s_{\min}(t)$ is fully annealed to $r/R$ (which is usually very small, e.g., $r/R = 0.125$ for a typical $64^2$ patch-wise training on $512^2$ resolution). As we show, the first issue makes the generator converge slower, and the second one makes the overall optimization less stable.

To mitigate this, we propose a small change in the pipeline by simply replacing the uniform scale sampling distribution with:

$$s \sim \text{Beta}(1, \beta(t)) \cdot (1 - r/R) + r/R, \qquad (5)$$

where $\beta(t)$ is gradually annealed from $\beta(0)$ to some final value $\beta(T)$. Using beta distribution instead of the uniform one gives a very convenient knob to shift the training focus between large patch scales $s \to 1$ (carrying the global information about the whole image) and small patch scales $r \to r/R$ (representing high-resolution local crops).

A natural way to do the annealing is to anneal from 0 to 1: at the start, the model focuses entirely on the structure, while at the end, it transforms into the uniform distribution (See Fig 4). We follow this strategy, but from the design perspective, set $\beta(T)$ to a value that is slightly smaller than 1 (we use $\beta(T) = 0.8$ everywhere) to keep more focus on the structure at the end of the annealing as well. In our initial experiments, $\beta(T) \in [0.7, 1]$ performs similarly. The scales distributions comparison between beta and uniform sampling is provided in Fig 4 and the convergence comparison in Fig 7.

### 3.4 Training details

We inherit the training procedure from StyleGAN2-ADA [24] with minimal changes. The optimization is performed by Adam [27] with a learning rate of 0.002 and betas of 0 and 0.99 for both G and D. We use $\beta(T) = 0.8$ for $T = 10000$, $z \sim \mathcal{N}(0, I)$ and set $R_p = 512$. D is trained with R1 regularization [37] with $\gamma = 0.05$. We train with the overall batch size of 64 for $\approx 15$M images seen by D for $256^2$ resolution and $\approx 20$M for $512^2$. Similar to previous works [6, 12], we use pose supervision for D for the FFHQ and Cats dataset to avoid geometry ambiguity. For this, we take the rotation and elevation angles, encode them with positional embeddings [59, 65] and feed them into a 2-layer MLP. After that, we multiply the obtained vector with the last hidden representation in the discriminator, following the Projection GAN [40] strategy from StyleGAN2-ADA [24]. We train G in full precision and use mixed precision for D. Since FFHQ has too noticeable 3D biases, we use generator pose conditioning for it [6]. Further details can be found in the source code.

## 4 Experiments

### 4.1 Experimental setup

**Benchmarks**. In our study, we consider four benchmarks: 1) FFHQ [25] in $256^2$ and $512^2$ resolutions, consisting of 70,000 (mostly front-view) human face images; 2) Cats $256^2$ [77], consisting of 9,998

---

[1] $\texttt{lerp}(x, y, \alpha) = (1 - \alpha) \cdot x + \alpha \cdot y$ for $x, y \in \mathbb{R}$ and $\alpha \in [0, 1]$.

[2] In practice, those methods use a *very* slightly different distribution (see Appx B)

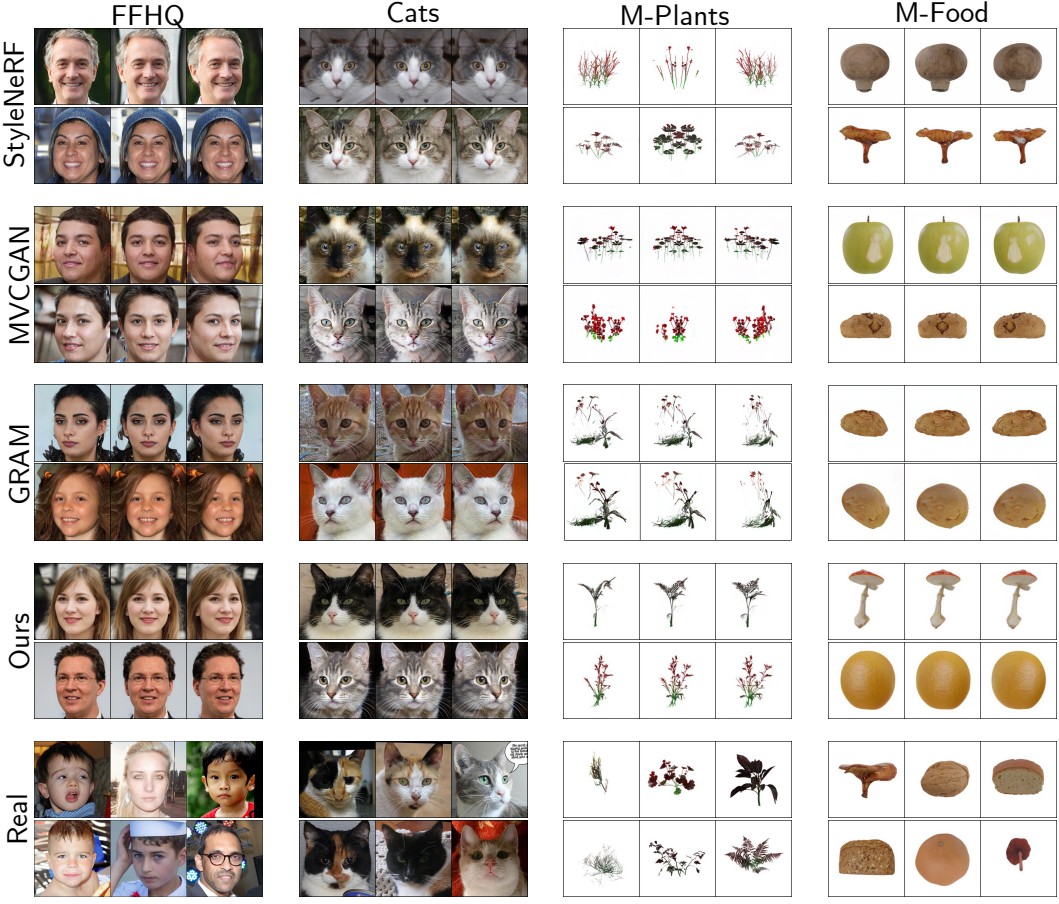

Figure 5: Comparing samples of EpiGRAF and modern 3D-aware generators. Our method attains state-of-the-art image quality, recovers high-fidelity geometry and preserves multi-view consistency for both simple-shape (FFHQ and Cats) and variable-shape (M-Plants and M-Food) datasets. We refer the reader to the supplementary for the video comparisons to evaluate multi-view consistency.

(mostly front-view) cat face images; 3) Megascans Food (M-Food) $256^2$ consisting of 199 models of different food items with 128 views per model (25472 images in total); and 4) Megascans Plants (M-Plants) $256^2$ consisting of 1108 different plant models with 128 views per model (141824 images in total). The last two datasets are introduced in our work to fix two issues with the modern 3D generation benchmarks. First, existing benchmarks have low variability of global object geometry, focusing entirely on a single class of objects, like human/cat faces or cars, that do not vary much from instance to instance. Second, they all have limited camera pose distribution: for example, FFHQ [25] and Cats [77] are completely dominated by the frontal and near-frontal views (see Appx E). That's why we obtain and render 1307 Megascans models from Quixel, which are photo-realistic (barely distinguishable from real) scans of real-life objects with complex geometry. Those benchmarks and the rendering code will be made publicly available.

**Metrics**. We use FID [20] to measure image quality and estimate the training cost for each method in terms of NVidia V100 GPU days needed to complete the training process.

**Baselines**. For upsampler-based baselines, we compare to the following generators: StyleNeRF [15], StyleSDF [47], EG3D [6], VolumeGAN [71], MVCGAN [78] and GIRAFFE-HD [72]. Apart from that, we also compare to pi-GAN [7] and GRAM [12], which are non-upsampler-based GANs. To compare on Megascans, we train StyleNeRF, MVCGAN, pi-GAN, and GRAM from scratch using their official code repositories (obtained online or requested from the authors), using their FFHQ or CARLA hyperparameters, except for the camera distribution and rendering settings. We also train StyleNeRF, MVCGAN, and $\pi$-GAN on Cats $256^2$. GRAM [12] restricts the sampling space to a set of learnable iso-surfaces, which makes it not well-suited for datasets with varying geometry.

Table 1: FID scores of modern 3D GANs. "†" — evaluated on a re-aligned version of FFHQ (different from original FFHQ [25]). Training cost is measured in terms of NVidia V100 GPU days. "OOM" denotes out-of-memory error.

| Method | FFHQ $256^2$ | FFHQ $512^2$ | Cats $256^2$ | M-Plants $256^2$ | M-Food $256^2$ | Training cost $256^2$ | Training cost $512^2$ | Geometry constraints |
|---|---|---|---|---|---|---|---|---|
| StyleNeRF [15] | 8.00 | 7.8 | 5.91 | 19.32 | 16.75 | 40 | 56 | $32^2$-res + 2D upsampler |
| StyleSDF [47] | 11.5 | 11.19 | – | – | – | 42 | 56 | $64^2$-res + 2D upsampler |
| EG3D [6] | 4.8† | 4.7† | – | – | – | N/A | 76 | $128^2$-res + 2D upsampler |
| VolumeGAN [71] | 9.1 | – | – | – | – | N/A | N/A | $64^2$-res + 2D upsampler |
| MVCGAN [78] | 13.7 | 13.4 | 39.16 | 31.70 | 29.29 | 42 | 64 | $64^2$-res + 2D upsampler |
| GIRAFFE-HD [72] | 11.93 | – | 12.36 | – | – | N/A | N/A | $16^2$-res + 2D upsampler |
| pi-GAN [7] | 53.2 | OOM | 68.28 | 75.64 | 51.99 | 56 | $\infty$ | none |
| GRAM [12] | 13.78 | OOM | 13.40 | 188.6 | 178.9 | 56 | $\infty$ | iso-surfaces |
| EpiGRAF (ours) | 9.71 | 9.92 | 6.93 | 19.42 | 18.15 | 16 | 24 | none |

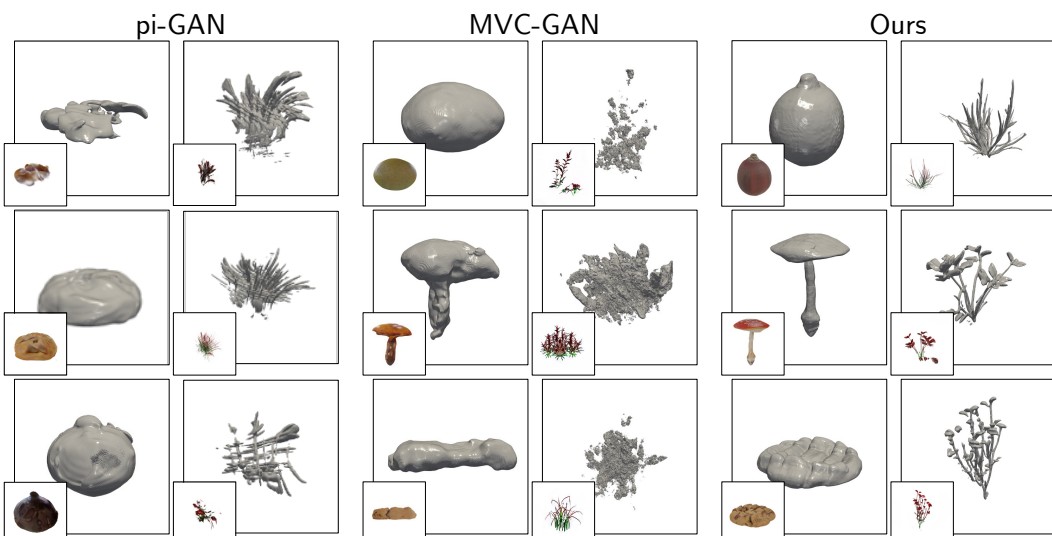

Figure 6: Visualizing the learned geometry for different methods. $\pi$-GAN [7] recovers high-fidelity shapes, but has worse image quality (see Table 1) and is much more expensive to train than our model. MVC-GAN [78] fails to capture good geometry because of the 2D upsampler. Our method learns proper geometry and achieves state-of-the-art image quality. We extracted the surfaces using marching cubes from the density fields sampled on $256^3$ grid and visualized them in PyVista [63]. We manually optimized the marching cubes contouring threshold for each checkpoint of each method. We noticed that $\pi$-GAN [7] produces a lot of "spurious" density which makes.

## 4.2 Results

*EpiGRAF achieves state-of-the-art image quality*. For Cats $256^2$, M-Plants $256^2$ and M-Food $256^2$, EpiGRAF outperforms all the baselines in terms of FID except for StyleNeRF, performing very similar to it on all the datasets even though it does not have a 2D upsampler. For FFHQ, our model attains very similar FID scores as the other methods, ranking 4/9 (including older $\pi$-GAN [7]), noticeably losing only to EG3D [6], which trains and evaluates on a different version of FFHQ and uses pose conditioning in the generator (which potentially improves FID at the cost of multi-view consistency). We provide a visual comparison for different methods in Fig 5.

*EpiGRAF is much faster to train*. As reported in Tab 1, existing methods typically train for $\approx$1 week on 8 V100s, EpiGRAF finishes training in just 2 days for $256^2$ and 3 days for $512^2$ resolutions, which is $2-3\times$ faster. Note that this high training efficiency is achieved without using an upsampler, which initially enabled the high-resolution synthesis of 3D-aware GANs. As to the non-upsampler methods, we couldn't train GRAM or $\pi$-GAN on $512^2$ resolution due to the memory limitations of the setup with 8 NVidia V100 32GB GPUs (i.e., 256GB of GPU memory in total).

*EpiGRAF learns high-fidelity geometry.* Using a pure NeRF-based backbone has two crucial benefits: it provides multi-view consistency and allows learning the geometry in the full dataset resolution. In Fig 6, we visualize the learned shapes on M-Food and M-Plants for 1) $\pi$-GAN: a pure NeRF-based generator without the geometry constraints; 2) MVC-GAN [78]: an upsampler-based generator with strong multi-view consistency regularization; 3) our model. We provide the details and analysis in the caption of Fig 6. We also provide the geometry comparison with EG3D on FFHQ $512^2$ in Fig 2.

*EpiGRAF easily capitalizes on techniques from the NeRF literature.* Since our generator is purely NeRF based and renders images without a 2D upsampler, it is well coupled with the existing techniques from the NeRF scene reconstruction field. To demonstrate this, we adopted background separation from NeRF++ [76] using the inverse sphere parametrization by simply copy-pasting the corresponding code from their repo. We depict the results in Fig 1 and provide the details in Appx B.

### 4.3 Ablations

We report the ablations for different discriminator architectures and patch sizes on FFHQ $512^2$ and M-Plants $256^2$ in Tab 2. Using a traditional discriminator architecture results in $\approx 15\%$ worse performance. Using several ones (via the group-wise convolution trick [26]) results in a noticeably slower training time and dramatically degrades the image quality. We hypothesize that the reason for it was the reduced overall training signal each discriminator receives, which we tried to alleviate by increasing their learning rate, but that did not improve the results. A too-small patch size hampers the learning process and produces a $\approx 80\%$ worse FID. A too-large one provides decent image quality but greatly reduces the training speed. Using a single scale/position-aware discriminator achieves the best performance, outperforming the standard one by $\approx 15\%$ on average.

To assess the convergence of our proposed patch sampling scheme, we compared against uniform sampling on Cats $256^2$ for $T \in \{1000, 5000, 10000\}$, representing different annealing speeds. We show the results for it in Fig 7: our proposed beta scale sampling strategy with $T = 10k$ schedule robustly converges to lower values than the uniform one with $T = 5k$ or $T = 10k$ and does not fluctuate much compared to the $T = 1k$ uniform one (where the model reached its final annealing stage in just 1k kilo-images seen by D).

To analyze how hyper-modulation manipulates the convolutional filters of the discriminator, we visualize the modulation weights $\sigma$, predicted by H, in Fig 8 (see the caption for the details). These visualizations show that some of the filters are always switched on, regardless of the patch scale; while others are always switched off, providing potential room for pruning [18]. And $\approx 40\%$ of the filters are getting switched on and off depending on the patch scale, which shows that H indeed learns to perform meaningful modulation.

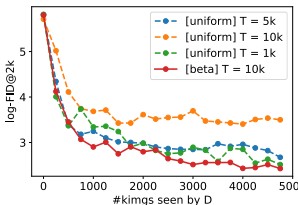

Figure 7: Convergence comparison on Cats $256^2$ for different sampling strategies.

| Experiment | FFHQ $512^2$ | M-Plants $256^2$ | Training cost |
|---|---|---|---|
| GRAF (with tri-planes) | 13.41 | 24.99 | 24 |
| + beta scale sampling ($T = 5k$) | 11.57 | 21.77 | 24 |
| + 2 scale-specific D-s | 10.87 | 21.02 | 28 |
| + 4 scale-specific D-s | 21.56 | 43.11 | 28 |
| + 1 scale/position-aware D | 9.92 | 19.42 | 24 |
| $- 32^2$ patch resolution | 17.44 | 34.32 | 19 |
| $- 64^2$ patch resolution (default) | 9.92 | 19.42 | 24 |
| $- 128^2$ patch resolution | 11.36 | 18.90 | 34 |

Table 2: Ablating the discriminator architecture and patch sizes in terms of FID scores and training cost (V100 GPU days).

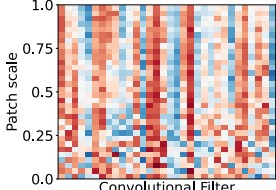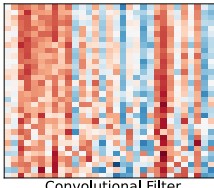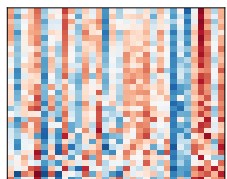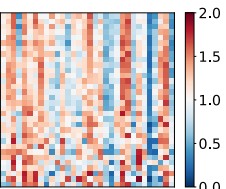

Figure 8: Visualizing modulation weights $\sigma$, predicted by H for 2-nd, 6-th, 10-th and 14-th convolutional layers. Each subplot denotes a separate layer and we visualize random 32 filters for it.

# 5    Limitations

**Performance drop for 2D generation**. Before switching to training 3D-aware generators, we spent a considerable amount of time, exploring our ideas on top of StyleGAN2 [24] for traditional 2D generation since it is faster, less error-prone and more robust to a hyperparameters choice. What we observed is that despite our best efforts (see C) and even with longer training, we couldn't obtain the same image quality as the full-resolution StyleGAN2 generator.

Table 3: Trying to train a traditional StyleGAN2 [26] generator in the patch-wise fashion. We tried to train longer to compensate for a smaller learning signal overall (a $64^2$ patch is $1/64$ of information compared to a $512^2$ image), but this didn't allow to catch up. Note, however, that AnyResGAN [5] reaches SotA when training on $256^2$ patches compared to $1024^2$ images.

| Method | FFHQ $512^2$ | | LSUN Bedroom $256^2$ | |
| --- | --- | --- | --- | --- |
| | FID | Training cost | FID | Training cost |
| StyleGAN2-ADA [24] | 3.83 | 8 | 4.12 | 5 |
| + multi-scale $64^2$ patch-wise training | 7.11 | 6 | 6.73 | 4 |
| + ×2 longer training | 5.71 | 12 | 5.42 | 8 |
| + ×4 longer training | 4.76 | 24 | 4.31 | 16 |

**A range of possible patch sizes is restricted**. Tab 2 shows the performance drop when using the $32^2$ patch size instead of the default $64^2$ one without any dramatic improvement in speed. Trying to decrease it further would produce even worse performance (imagine training in the extreme case of $2^2$ patches). Increasing the patch size is also not desirable since it decreases the training speed a lot: going from $64^2$ to $128^2$ resulted in 30% cost increase without clear performance benefits. In this way, we are very constrained in what patch size one can use.

**Discriminator does not see the global context**. When the discriminator classifies patches of small scale, it is forced to do so without relying on the global image information, which could be useful for this. Our attempts to incorporate it (see Appx C) did not improve the performance (though we believe we under-explored this).

**Low-resolution artifacts**. While our generator achieves good FID on FFHQ $512^2$, we noticed that it has some blurriness when one zooms-in into the samples. It is not well captured by FID since it always resizes images to the $299 \times 299$ resolution. We attribute this problem to our patch-wise training scheme, which puts too much focus on the structure and believe that it could be resolved.

# 6    Conclusion

In this work, we showed that it is possible to build a state-of-the-art 3D GAN framework without a 2D upsampler, but using a pure NeRF-based generator trained in a multi-scale patch-wise fashion. For this, we improved the traditional patch-wise training scheme in two important ways. First, we proposed to use a scale/location-aware discriminator with convolutional filters modulated by a hypernetwork depending on the patch parameters. Second, we developed a schedule for patch scale sampling based on the beta distribution, that leads to faster and more robust convergence. We believe that the future of 3D GANs is a combination of efficient volumetric representations, regularized 2D upsamplers, and patch-wise training. We propose this avenue of research for future work.

Our method also has several limitations. Before switching to training 3D-aware generators, we spent a considerable amount of time exploring our ideas on top of StyleGAN2 for traditional 2D generation, which always resulted in higher FID scores. Further, the discriminator loses information about global context. We tried multiple ideas to incorporate global context, but it did not lead to an improvement. Next, our current patch-wise training scheme might cause some low-res artifacts. Finally, 3D GANs generating faces and humans may have negative societal impact as discussed in Appx H.

# 7    Acknowledgements

We would like to acknowledge support from the SDAIA-KAUST Center of Excellence in Data Science and Artificial Intelligence.

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
