## A    Additional limitations due to aliasing

Current patch-wise training strategies (both ours and in prior works [56, 36]) do not take aliasing into account when extracting patches. This leads to additional problems, which we illustrate in Figure 9. Basically, sampling patches from images (when performed naively) is prone to aliasing, which can potentially result in learning an incorrect distribution.

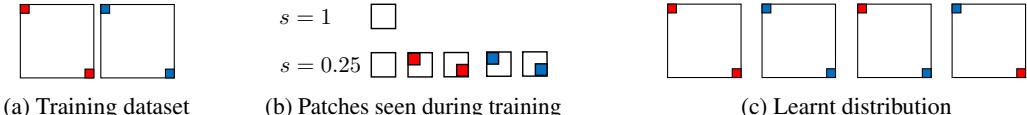

(a) Training dataset         (b) Patches seen during training         (c) Learnt distribution

Figure 9: Illustrating fundamental limitations of a non-anti-aliased patch-wise training scheme on a toy example. Consider a simple dataset consisting of just two images, illustrated on Figure 9a, which are mostly white except for the colored corners. Consider a generator trained with the patch-wise scheme without taking anti-aliasing into account, where we sample patches of just two scales: $s = 1$ and $s = 0.25$. Then, during training it would only see patches as depicted in Figure 9b, where color correlation in the corners would not be noticeable since large-scale patches will lose such find-grained details during aliased down-sampling (as in both our and previous patch-wise training schemes [56, 36]). As a result, the generator will learn to generate images without taking correlations between high-frequency details into account, as illustrated in Figure 9c.

We can put this more rigorously. ' Imagine that we have a distribution $p(\boldsymbol{x})$ for $\boldsymbol{x} \in \mathbb{R}^{R \times R}$, where we consider images to be single-channel (for simplicity) and $R$ is the image size. If we train with patch size of $r$, then each optimization step uses random $r \times r$ pixels out of $R \times R$, i.e. we optimize over the distribution of all possible marginals $p(\boldsymbol{x}_p)$ where $\boldsymbol{x}_p = e(\boldsymbol{x}; \xi) \in \mathbb{R}^{r \times r}$ is an image patch and $e(\boldsymbol{x}, \xi)$ is an aliased patch extraction function with nearest neighbor interpolation and random seed $\xi$.[3] This means that our minimax objective becomes:

$$\min_{\mathsf{G}} \max_{\mathsf{D}} \mathbb{E}_{p(\boldsymbol{x}_p)}[\log \mathsf{D}(\boldsymbol{x}_p)] + \mathbb{E}_{p(\boldsymbol{z}), p(\xi)}[\log(1 - \mathsf{D}(e(\mathsf{G}(\boldsymbol{z}), \xi)))] \tag{6}$$

If we rely on the GAN convergence theorem [14], stating that we recover the training distribution as the solution of the minimax problem, then G will learn to approximate all the possible marginal distributions $p(\boldsymbol{x}_p)$ instead of the full joint distribution $p(\boldsymbol{x})$, that we seek.

Sampling patches in an alias-free manner is tricky for our generator, since we cannot obtain intermediate high-resolution representations (which are needed to address aliasing) from tri-planes due to the computational overhead. We leave the development of patch sampling schemes for NeRF-based generators for future work.

## B    Training details

### B.1    Hyper-parameters and optimization details

We inherit most of the hyperparameters from the StyleGAN2-ADA repo [24] repo which we build on top[4]. In this way, we use the dimensionalities of $512$ for both $\boldsymbol{z}$ and $\boldsymbol{w}$. The mapping network has 2 layers of dimensionality 512 with LeakyReLU non-linearities with the negative slope of $-0.2$. Synthesis network S produced three $512^2$ planes of 32 channels each. We use the SoftPlus non-linearity instead of typically used ReLU [38] as a way to clamp the volumetric density. Similar to $\pi$-GAN, we also randomize

For FFHQ and Cats, we also use camera conditioning in D. For this, we encode yaw and pitch angles (roll is always set to 0) with Fourier positional encoding [59, 65], apply dropout with 0.5 probability (otherwise, D can start judging generations from 3D biases in the dataset, hurting the image quality), pass through a 2-layer MLP with LeakyReLU activations to obtain a 512-dimensional vector, which is finally as a projection conditioning [40]. Cameras positions were extracted in the same way as in GRAM [12].

---

[3]For brevity, we "hide" all the randomness of the patch sampling process into $\xi$.
[4]https://github.com/NVlabs/stylegan2-ada-pytorch

We optimize both G and D with the batch size of 64 until D sees 25,000,000 real images, which is the default setting from StyleGAN2-ADA. We the default setup of adaptive augmentations, except for random horizontal flipping, since it would require the corresponding change in the yaw angle at augmentation time, which was not convenient to incorporate from the engineering perspective. Instead, random horizontal flipping is used non-adaptively as a dataset mirroring where flipping the yaw angles is more accessible. We train G in full precision, while D uses mixed precision.

Hypernetwork H is structured very similar to the generator's mapping network. It consists of 2 layers with LeakyReLU non-linearities with the negative slope of $-0.2$. Its input is the positional embedding of the patch scales and offsets $s, \delta_x, \delta_y$, encoded with Fourier features [59, 65] and concatenated into a single vector of dimensionality 828. It produces a patch representation vector $\boldsymbol{p} \in \mathbb{R}^{512}$, which is then adapted for each convolutional layer via:

$$\boldsymbol{\sigma} = \tanh(\boldsymbol{W}_\ell \boldsymbol{p} + \boldsymbol{b}_\ell) + 1, \tag{7}$$

where $\boldsymbol{\sigma} \in [0, 2]^{c_{\text{out}}^\ell}$ is the modulation vector, $(\boldsymbol{W}_\ell, \boldsymbol{b}_\ell)$ is the layer-specific affine transformation, $c_{\text{out}}^\ell$ is the amount of output filters in the $\ell$-th layer. In this way, H has layer-specific adapters.

For the background separation experiment, we adapt the neural representation MLP from INR-GAN [60], but passing 4 coordinates (for the inverse sphere parametrization [76]) instead of 2 as an input. It consists of 2 blocks with 2 linear layers each. We use 16 steps per ray for the background without hierarchical sampling.

Further details could be found in the accompanying source code.

### B.2  Utilized computational resources

While developing our model, we had been launching experiments on $4\times$ NVidia A100 81GB or Nvidia V100 32GB GPUs with the AMD EPYC 7713P 64-Core processor. We found that in practice, running the model on A100s gives a $2\times$ speed-up compared to V100s due to the possibility of increasing the batch size from 32 to 64. In this way, training EpiGRAF on $4\times$ A100s gives the same training speed as training it $8\times$ V100s.

For the baselines, we were running them on 4-8$\times$ V100s GPUs as was specified by the original papers unless the model could fit into 4 V100s without decreasing the batch size (it was only possible for StyleNeRF [15]).

For rendering Megascans, we used $4\times$ NVIDIA TITAN RTX with 24GB memory each. But resource utilization for rendering is negligible compared to training the generators.

In total, the project consumed $\approx$4 A100s GPU-years, $\approx$4 V100s GPU-years, and $\approx$20 TITAN RTX GPU-days. Note, that out of this time, training the baselines consumed $\approx$1.5 V100s GPU-years.

### B.3  Annealing schedule details

As being said in §3.3, the existing multi-scale patch-wise generators [56, 36] use uniform distribution $U[s_{\min(t)}, 1]$ to sample patch scales, where $s_{\min(t)}$ is gradually annealed during training from 0.9 (or 0.8 [36]) to $r/R$ with different speeds. We visualize the annealing schedule for both GRAF and GNeRF on Fig 10, which demonstrates that their schedules are very close to `lerp`-based one, described in §3.3.

## C  Failed experiments

Modern GANs are a lot of engineering and it often takes a lot of futile experiments to get to a point where the obtained performance is acceptable. We want to enumerate some experiments which did not work out (despite looking like they should work) — either because the idea was fundamentally flawed on its own or because we've under-explored it (or both).

**Conditioning D on global context worsened the performance**. In Appx 5, we argued that when D processes a small-scale patch, it does not have access to the global image information, which might be a source of decreased image quality. We tried several strategies to compensate for this. Our first attempt was to generate a low-resolution image, bilinearly upsample it to the target size, and then

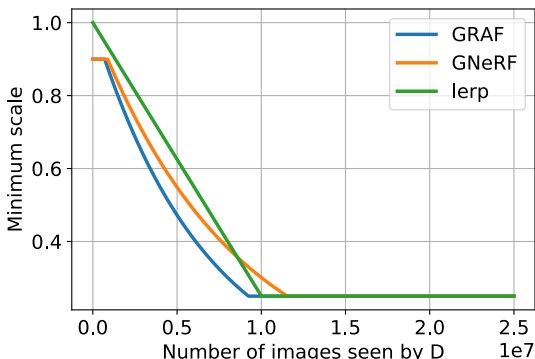

Figure 10: Comparing annealing schedules for GRAF [56], GNeRF [36] and the `lerp`-based schedule from §3.3. We simplified the exposition by stating that GRAF and GNeRF use the `lerp`-based schedule, which is *very* close to reality.

"grid paste" a high-resolution patch into it. The second attempt was to simply always concatenate a low-resolution version of an image as 3 additional channels. However, in both cases, generator learned to produce low-resolution version of images well, but the texture was poor. We hypothesize that it was due to D starting to produce its prediction almost entirely based on the low-resolution image, ignoring the high-resolution patches since they are harder to discriminate.

**Patch importance sampling did not work**. Almost all the datasets used for 3D-aware image synthesis have regions of difficult content and regions with simpler content — it is especially noticeable for CARLA [56] and our Megascans datasets, which contain a lot of white background. That's why, patch-wise sampling could be improved if we sample patches from the more difficult regions more frequently. We tried this strategy in the GNeRF [36] problem setup on the NeRF-Synthetic dataset [38] of fitting a scene without known camera parameters. We sampled patches from regions with high average gradient norm more frequently. For some scenes, it helped, for other ones, it worsened the performance.

**View direction conditioning breaks multi-view consistency**. Similar to the prior works [56, 7], our attempt to condition the radiance (but not density) MLP on ray direction (similar to NeRF [38]) led to poor multi-view consistency with radiance changing with camera moving. We tested this on FFHQ [25], which has only a single view per object instance and suspect that it wouldn't be happening on Megascans, where view coverage is very rich.

**Tri-planes produced from convolutional layers are much harder to optimize for reconstruction**. While debugging our tri-plane representation, we found that tri-planes produced with convolutional layers are extremely difficult to optimize for reconstruction. I.e., if one fits a 3D scene while optimizing tri-planes directly, then everything goes smoothly, but when those tri-planes are being produced by the synthesis network of StyleGAN2 [26], then PNSR scores (and the loss values) are plateauing very soon.

## D Additional patch size ablation

Table 2 shows that the generator achieves the best performance for the $64^2$ patch resolution. The comparison between patch sizes was performed while keeping all other hyperparameters fixed. This creates an issue since StyleGAN-based generators should use different values for the R1 regularization weight $\gamma$ depending on the training resolutions.[5] This is why in Table 4, we provide the results for a $3 \times 3$ grid search over patch sizes and R1 regularization gamma.

And this better aligns with intuition: increasing the patch size should improve the performance (at the loss of the training speed) since the model uses more information during training. The main reason why we fixed the patch resolution to $64^2$ is because we considered the computational overhead not to be worth the quality improvements it brings: while it is not expensive to run several individual

---

[5]See https://github.com/NVlabs/stylegan2-ada-pytorch/blob/main/train.py#L173.

Table 4: FID@2k scores for the grid search over the patch size and R1 regularization weight $\gamma$ on Cats $256^2$. Each model was trained for 5M seen images and FID was measured using 2048 fake and all the real images.

| Patch size | $\gamma = 0.01$ | $\gamma = 0.1$ | $\gamma = 1$ | Training speed |
|---|---|---|---|---|
| $32^2$ | 20.31 | 30.72 | 335.32 | 9.84 seconds / 1K images |
| $64^2$ | 24.93 | 18.13 | 20.07 | 11.36 seconds / 1K images |
| $128^2$ | 21.21 | 18.72 | 16.96 | 17.45 seconds / 1K images |

experiments with the $128^2$ patch resolution, it is expensive to develop the whole project around the $128^2$ patch resolution generator.

# E  Datasets details

## E.1  Megascans dataset

Modern 3D-aware image synthesis benchmarks have two issues: 1) they contain objects of very similar global geometry (like, human or cat faces, cars and chairs), and 2) they have poor camera coverage. Moreover, some of them (e.g., FFHQ), contain 3D-biases, when an object features (e.g., smiling probability, gaze direction, posture or haircut) correlate with the camera position [6]. As a result, this does not allow to evaluate a model's ability to represent the underlying geometry and makes it harder to understand whether performance comes from methodological changes or better data preprocessing.

To mitigate these issues, we introduce two new datasets: Megascans Plants (M-Plants) and Megascans Food (M-Food). To build them, we obtain $\approx 1,500$ models from Quixel Megascans[6] from Plants, Mushrooms and Food categories. Megascans are very high-quality scans of real objects which are almost indistinguishable from real. For Mushrooms and Plants, we merge them into the same Food category since they have too few models on their own.

We render all the models in Blender [3] with cameras, distributed uniformly at random over the sphere of radius 3.5 and field-of-view of $\pi/4$. While rendering, we scale each model into $[-1, 1]^3$ cube and discard those models, which has the dimension produce of less than 2. We render 128 views per object from a fixed distance to the object center from uniformly sampled points on the entire sphere (even from below). For M-Plants, we additionally remove those models which have less than 0.03 pixel intensity on average (computed as the mean alpha value over the pixels and views). This is needed to remove small grass or leaves which will be occupying a too small amount of pixels. As a result, this procedure produces 1,108 models for the Plants category and 199 models for the Food category.

We include the rendering script as a part of the released source code. We cannot release the source models or textures due to the copyright restrictions. We release all the images under the CC BY-NC-SA 4.0 license[7]. Apart from the images, we also release the class categories for both M-Plants and M-Food.

The released datasets do not contain any personally identifiable information or offensive content since it does not have any human subjects, animals or other creatures with scientifically proven cognitive abilities. One concern that might arise is the inclusion of Amanita muscaria[8] into the Megascans Food dataset, which is poisonous (when consumed by ingestion without any specific preparation). This is why we urge the reader not to treat the included objects as edible items, even though they are a part of the "food" category. We provide random samples from both of them in Fig 11 and Fig 12. Note that they are almost indistinguishable from real objects.

---

[6]https://quixel.com/megascans
[7]https://creativecommons.org/licenses/by-nc-sa/4.0
[8]https://en.wikipedia.org/wiki/Amanita_muscaria

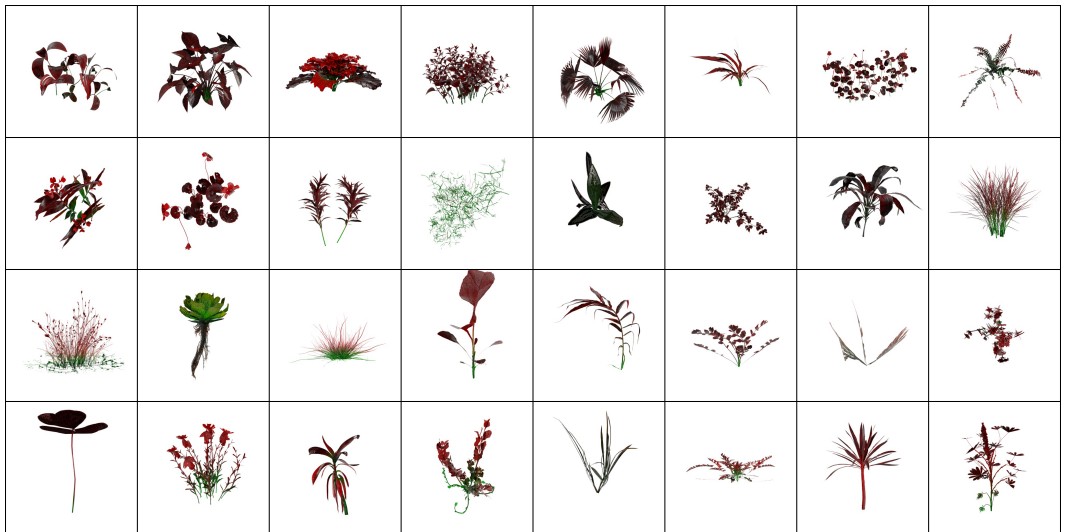

Figure 11: Real images from the Megascans Plants dataset. This dataset contains very complex geometry and texture, while having good camera coverage.

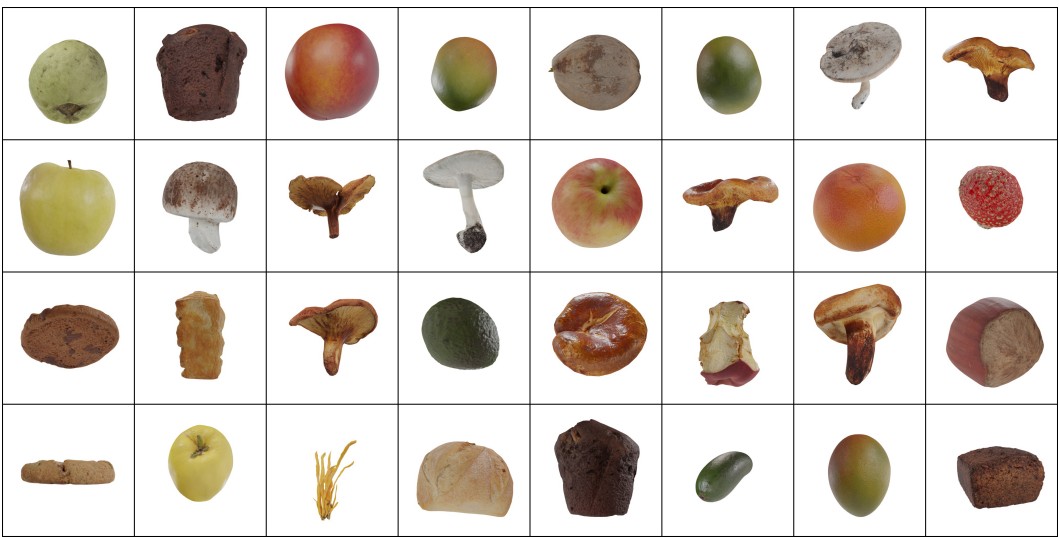

Figure 12: Real images from the Megascans Food dataset. Caution: some objects in this dataset could be poisonous.

Table 5: Comparing 3D datasets. Megascans Plants and Megascans Food are much more complex in terms of geometry and has much better camera coverage than FFHQ [25] or Cats [77]. The abbreviation "USphere($\mu, \zeta$)" denotes uniform distribution on a sphere (see $\pi$-GAN [7]) with mean $\mu$ and pitch interval of $[\mu - \zeta, \mu + \zeta]$. For Cats, the final resolution depends on the cropping and we report the original dataset resolution.

| Dataset | Number of images | Yaw distribution | Pitch distribution | Resolution |
|---------|------------------|------------------|--------------------|------------|
| FFHQ [25] | 70,000 | Normal(0, 0.3) | Normal($\pi/2$, 0.2) | $1024^2$ |
| Cats | 10,000 | Normal(0, 0.2) | Normal($\pi/2$, 0.2) | $\approx 604 \times 520$ |
| CARLA | 10,000 | USphere(0, $\pi$) | USphere($\pi/4$, $\pi/4$) | $512^2$ |
| M-Plants | 141,824 | USphere(0, $\pi$) | USphere($\pi/2$, $\pi/2$) | $1024^2$ |
| M-Food | 25,472 | USphere(0, $\pi$) | USphere($\pi/2$, $\pi/2$) | $1024^2$ |

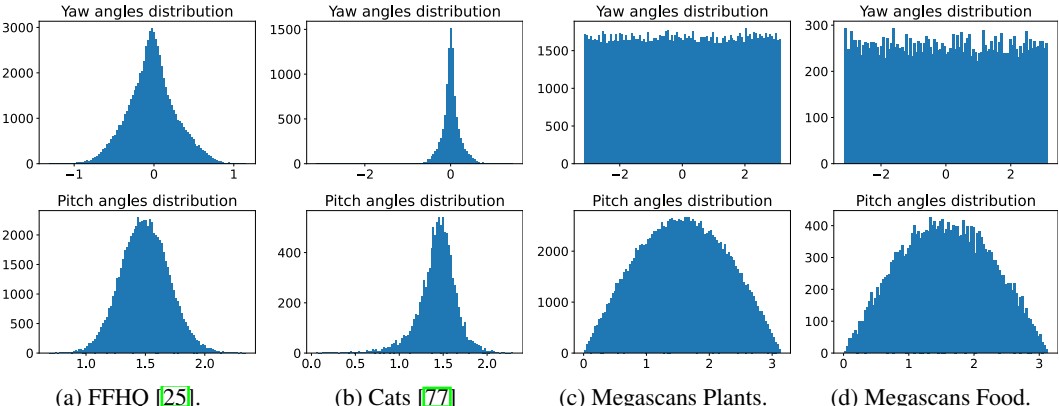

(a) FFHQ [25].     (b) Cats [77]     (c) Megascans Plants.     (d) Megascans Food.

Figure 13: Comparing yaw/pitch angles distribution for different datasets.

### E.2 Datasets statistics

We provide the datasets statistics in Tab 5. For CARLA [56], we provide them for comparison and do not use this dataset as a benchmark since it is small, has simple geometry and texture.

## F Additional samples

We provide random non-cherry-picked samples from our model in Fig 14, but we recommend visiting the website for video illustrations: https://universome.github.io/epigraf.

To demonstrate GRAM's [12] mode collapse, we provide more its samples in Fig 15.

## G Potential negative societal impacts

Our developed method is in the general family of media synthesis algorithms, that could be used for automatized creation and manipulation of different types of media content, like images, videos or 3D scenes. Of particular concern is creation of deepfakes[9] — photo-realistic replacing of one person's identity with another one in images and videos. While our model does not yet rich good enough quality to have perceptually indistinguishable generations from real media, such concerns should be kept in mind when developing this technology further.

---

[9]https://en.wikipedia.org/wiki/Deepfake

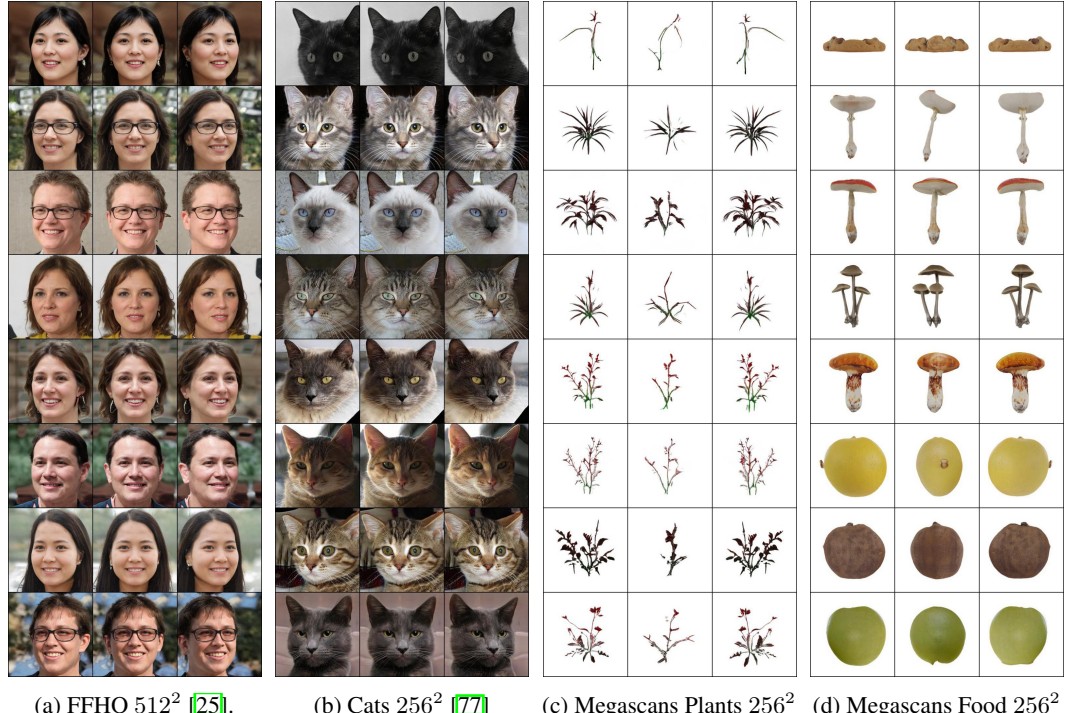

(a) FFHQ $512^2$ [25].     (b) Cats $256^2$ [77]     (c) Megascans Plants $256^2$     (d) Megascans Food $256^2$

Figure 14: Random samples (without any cherry-picking) for our model. Zoom-in is recommended.

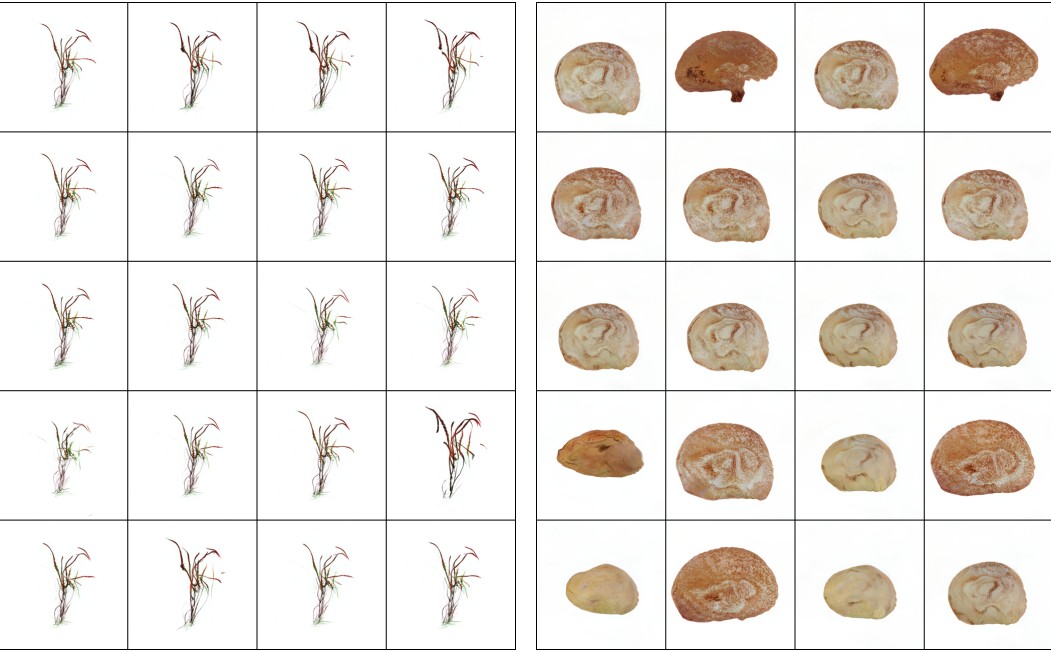

Figure 15: Random samples from GRAM [12] on M-Plants $256^2$ (left) and M-Food $256^2$ (right). Since it uses the same set of iso-surfaces for each sample to represent the geometry, it struggles to fit the datasets with variable structure, suffering from a very severe mode collapse.

## H    Ethical concerns

We have reviewed the ethics guidelines[10] and confirm that our work complies with them. As discussed in Appx E.1, our released datasets are not human-derived and hence do not contain any personally identifiable information and are not biased against any groups of people.

---

[10]https://nips.cc/public/EthicsGuidelines