# OpenReview forum: "EpiGRAF: Rethinking training of 3D GANs"
_NeurIPS.cc/2022/Conference — NeurIPS 2022 Accept_

### Official Review · Reviewer_bBXS · 2022-06-27

**Rating:** 5
**Confidence:** 3
**Soundness:** 3 good
**Presentation:** 4 excellent
**Contribution:** 2 fair

**Summary:**

The authors propose a framework to generate 3D images from latent code. Authors analyze in details the drawbacks of current methods, based on upsampling strategy that could affect multi-view consistency or rely on low resolution rendering. They came up with a patch based strategy similar to GRAF to improve the performance. The idea is to condition the discriminator with a hypernetwork to better make sense of the output of the convolutional filter at multiple scales. Additionally, they propose a beta distribution when sampling the scale, improving the overall convergence

**Questions:**

I'd like the authors to comment on the limitations I described in the Strength and Weakness section. In particular, why one could not simply rely on the EG3D triplane representation + GRAF patch based discriminator to achieve the same results proposed here. Is there any experiment showing that it's not sufficient?

Also, why one should prefer this method instead of EG3D? Seems to obtain better quantitative metrics (and qualitative metrics are not shown).

**Limitations:**

Yes, Limitations and Ethical concerns are described in the supplementary material

**Strengths And Weaknesses:**

+++ Relevance. The work is super relevant for the community, with a lot of possible applications.

+++ Clarity. The paper is very well written and well motivated. Authors describe in details all the aspects of 3D generators, explaining their drawbacks and how to mitigate those. In particular, I really appreciated in the supplementary material the attempts at 2D generation and all the failing experiments the authors reported. This could be very useful for the community as GAN methods usually rely on a substantial amount of trial and errors before getting to the right combination of architecture and loss functions.

--- Novelty and Experiments. Whereas this paper has many good aspects and I'd like to see it published, I have a few doubts regarding the overall novelty and the missing comparisons.

In particular, the method "borrows" multiple aspects from EG3D and GRAF, but comparisons are very limited. Indeed EG3D outperforms this method substantially in terms of FID, authors hint that the upsampler leads to a better FID score but worse multiview consistency (l240), but they do not provide any example of this.

Similarly, the method borrows the patch-based, multi-scale discriminator idea from GRAF, improving with the conditioning from a hypernetwork, but I am not sure there is a direct comparison with GRAF?

There are no examples of inferred geometry. Ideally authors would also show the inferred geometry and qualitatively compare with EG3D to prove that the method leads to better multi-view consistency.

---

> ### Author Response · Authors · 2022-08-05
> **Official authors' response [Part 2/2]**
>
> **Q**. *Multi-view consistency comparison with upsampler-based generators? With EG3D?*
>
> **A**.
> We provided video comparisons with upsampler-based methods in terms of multi-view consistency on our supplementary material website: [https://rethinking-3d-gans.github.io/](https://rethinking-3d-gans.github.io/).
> Unfortunately, showing a video is the only reasonable way to assess multi-view consistency, which we cannot include in the main text.
> Videos on the website for all four datasets (FFHQ, Cats, M-Plants and M-Food) show that StyleNeRF and MVC-GAN (SotA generators which had their code released at the time of the submission) have their texture/shapes changing when the camera moves. At the same time, the samples of our model remain consistent.
>
> Since the official source code for EG3D is out, we provide the comparisons to it in Section 2 on [this web page](https://rethinking-3d-gans.github.io/additional-results).
> One can still see that EG3D alters the texture (especially in the hair or around the eyes) and sometimes even structure (near the mouth) depending on the camera position.
> This lack of multi-view consistency is undesirable in practical applications since it looks like aliasing problems or flickering to the end user.
>
> =====================================================================
>
> **Q**. *Could you provide a qualitative comparison with EG3D in terms of inferred geometry?*
>
> **A**.
> That being said, we couldn't do this at the submission time since their codebase was not released.
> We provide this comparison on FFHQ $512^2$ in Section 1 on [this web page](https://rethinking-3d-gans.github.io/additional-results).
> We recommend viewing those videos in full resolution: one can see from them that our method captures high-frequency surface details (like hair or areas around the eyes) much better since it learns the shapes in the natural dataset resolution.
>
> =====================================================================
>
> **Q**. *Why could one not simply rely on the tri-plane representation + GRAF patch-based discriminator to achieve the same results proposed here? Is there a direct comparison with GRAF?*
>
> **A**.
> GRAF is a foundational model, but much time has passed since it was released, and directly comparing to it would be unfair since it uses older architectural backbones and trains on small compute.
> At the submission time, we did not include the comparison to the bare tri-planes + GRAF's uniform scale sampling setup and ablated each component (discriminator structure and beta sampling strategy) separately.
> But we agree that such a comparison would be helpful to a reader and run this model on FFHQ $512^2$, M-Plants $256^2$ for $T=5k$ and Cats $256^2$ for $T=1k, 5k, 10k$ and with the proposed beta scale sampling to compare the convergence for different scale sampling strategies for it.
> The bare tri-planes + GRAF sampling setup achieves ${\approx}30$% worse performance than our final model.
> We report the scores in Table 2 (see the updated submission) and the convergence plots comparison for different scale sampling strategies in Section 5 on [this web page](https://rethinking-3d-gans.github.io/additional-results) (and add it to the supplementary).
> We thank the reviewer for bringing this up: it helps us to improve our work.

---

> > ### Comment · Reviewer_bBXS · 2022-08-08
> > **Thank you for the rebuttal**
> >
> > I thank the authors for the exhaustive reply. I believe the response clarified my concerns and I am leaning towards acceptance.

---

> ### Author Response · Authors · 2022-08-05
> **Official authors' response [Part 1/2]**
>
> We would love to express our appreciation to the reviewer for their thorough analysis of our work and the raised concerns and questions — they allow us to make our submission stronger.
> In the following, we provide additional experimental results and visualizations, details on the questions and concerns, and elaborate on the advantages of our model.
>
> **Q**. *Why should one prefer this method instead of EG3D? It seems to obtain better quantitative metrics (and qualitative metrics are not shown).*
>
> **A**.
> First of all, note that EG3D's official codebase was released only after the NeurIPS submission deadline, that's why we could compare to it only in terms of the reported scores.
> Next, note that EG3D is trained not on the standard FFHQ but on a re-cropped/re-aligned version of it, that's why FID scores are not directly comparable for it with other 3D generators.
> Then, we would also want to note that EG3D is extremely well-tuned for FFHQ both in terms of sheer compute and engineering efforts: it contains a variety of advanced techniques which improve its performance and which we couldn't borrow since its official codebase was released after the submission deadline, like density regularization, some parts of Mip-NeRF volume rendering, richer information about camera poses (i.e., following prior work, we assume zero roll angles, while in their case they use all three rotation/elevation/roll angles), and others.
> Besides, our project spent ${\approx}$12 NVidia V100 GPU-years in total (where $1.5$ GPU-years were spent on running/tuning baselines), while just a single run of EG3D on FFHQ $512^2$ is ${\approx}0.2$ V100 GPU-years.
> While EG3D does not report their overall compute, the recent StyleGAN-s (StyleGAN2-ADA/StyleGAN3) typically consume 90-140 GPU-years per project, and it feels safe to assume that EG3D falls into a similar class of projects.
> And the amount of spent compute determines the possibility of finding better hyperparameters for a model and the number of ideas one can explore to improve the performance.
>
> Moreover, training EG3D on a new dataset seems to be not that straightforward: we attempted to train it on the standard FFHQ (to get directly comparable FID scores) with the official hyper-parameters provided for their re-aligned version of FFHQ and we observe considerably lower performance: for now, $40\%$ of its training has passed and its current FID is 24.5.
> For comparison, StyleGAN2 achieves an FID of 6.08 at this training stage, and our generator — 14.78.
> We include the samples from it in Section 7 on [this web page](https://rethinking-3d-gans.github.io/additional-results).
> In this way, it seems that EG3D needs some tuning when one applies it to a new dataset.
>
> Answering the original question, we can name the following reasons why our model could be preferable:
> - It is 3 times cheaper to train.
> - It is multi-view consistent by construction and captures high-frequency geometry details (see the visualizations in Section 1 and Section 2 on [this web page](https://rethinking-3d-gans.github.io/additional-results)).
> - Our model can easily integrate tricks from the existing NeRF literature: i.e., we did background separation on FFHQ by simply copy-pasting the code from NeRF++ (see Figure 1 of the main text).
>
> Finally, we believe that our paper is not simply a novel generator architecture (which is useful for its own sake) but also our exploration of patch-wise sampling and improvements we develop on top of it, which could be used in other scenarios.
> At the current moment, the community almost completely abandoned patch-wise training (even the original GRAF authors didn't use it in their recent [VoxGRAF project](https://katjaschwarz.github.io/voxgraf)) and switched to upsampler-based generators. However, in our work we show that patch-wise models are powerful rivals for them both in terms of performance and training cost, which we believe is important knowledge for the community to have.

---

### Official Review · Reviewer_LXyr · 2022-07-10

**Rating:** 6
**Confidence:** 4
**Soundness:** 4 excellent
**Presentation:** 4 excellent
**Contribution:** 3 good

**Summary:**

This paper proposes two important techniques to train 3D generative models from 2D supervision.

Existing works adopt nerf to render 3D radiance fields into 2D images with a large memory cost. As a result, they fail to train the discriminator with high resolution and more often render 2D images with low resolution and generate high-res images by 2D CNN up-sampling. However, it brings strong view-inconsistent problems.

This paper explores patched-based discriminator. By rendering small patches, the generator can directly synthesize high-res images without any upsampling tricks. To improve the quality of patch discriminator, the paper proposes 1) a location- and scale-aware discriminator and 2) a beta distribution patch sampling strategy, both showing good improvement. It also produces quite nice and view consistent results and outperforms the baselines.


**Questions:**

I have several questions about the implementation details.

1)	As for the rendering equation, did the authors adopt the original ray-based rendering equation (in original nerf paper), or light cone based one (in mip-nerf?) If I didn’t understand wrong, it seems the authors choose the former. In such a case, would it produce anti-aliasing problems when rendering a 64x64 patch with the scale to be 1(let’s say the original image is 256x256)? Adopting the mip-nerf seems to be the straightforward solution. Could the authors discuss this part?

2)	How do you make real patch data for training the discriminator?  Are you cropping the real images with specific scales, then resizing them to be 64x46? I wonder do you also consider the antialiasing problem here?

3)	What’s the relationship between patch size and final image size? E.g., In table2 it seems for 512x512 image size, 64 patch size seems to be the best. Does it mean that for 256x256 image size we should change to 32 patch size?

4)	It is good to see that the proposed generators can generate up to 512x512 images. I wonder what's maximum. Can it be scaled to 1024x1024?

5)	For face and cat, the camera poses are limited to the front views.  I wonder can you do large view changes, e.g., rendering side faces, or rendering from overhead? I want to know whether the radiance field will remain view-consistent for unseen training views. On the other side, I recognize that if the training images cover 360 degrees of the object, then the radiance field doesn’t have such issues.


**Ethics Review Area:**

["I don’t know"]

**Limitations:**

I appreciate that the authors honestly report that the proposed techniques don’t work in 2D cases. However, I treat it as a limitation of the paper since it undermines the generalization of the techniques.

**Strengths And Weaknesses:**

Strength:
The patch-based solution to solve the view-inconsistency problem of nerf-based generative models is reasonable and the two proposed techniques are quite simple but seem to work well.
The final FID scores are better than all the baselines with much less training time needed, and the results looks quite nice!

Weakness:
It is very weird why the strategy cannot be applied in 2D cases, which raises concerns of generalization.

---

> ### Author Response · Authors · 2022-08-05
> **Official authors' response [Part 2/2]**
>
> **Q**. *What's the relationship between patch size and final image size? E.g., In table2 it seems for 512x512 image size, 64 patch size seems to be the best. Does it mean that for 256x256 image size, we should change to 32 patch size?*
>
> **A**.
> We believe that, in general, the higher, the better, since higher-resolution patches contain more information and hence learning signal for both the generator and discriminator.
> Our project evolved around the highest affordable resolution in our resource constraints: $64^2$, and to be honest, we believe that the performance for the $128^2$ patch size could be improved if one tune it better.
> In our case, it was simply not affordable since the training takes too long (40\% more, which was a permissible cost for us for final ablations, but not to develop the whole project).
>
> To compare more thoroughly, we launched a hyper-parameter grid search on Cats $256^2$ for $r = 32^2, 64^2$ $128^2$ patch sizes and $\gamma = 0.01, 0.1$ and $1.0$ R1 penalty weight, which is the most important hyper-parameter when exploring new resolution or dataset for StyleGAN2-ADA (in our work, we used $\gamma=0.05$ which we inherited from our patch-wise training experiments on 2D generation).
> The FID@2k scores for them are provided below:
> - $32^2$ patch size:
>     - $\gamma=0.01$: 20.31
>     - $\gamma=0.1$: 30.72
>     - $\gamma=1$: 335.32
>     - training speed: 9.84 seconds / 1K images
> - $64^2$ patch size:
>     - $\gamma=0.01$: 24.93
>     - $\gamma=0.1$: 18.13
>     - $\gamma=1$: 20.07
>     - training speed: 11.36 seconds / 1K images
> - $64^2$ patch size:
>     - $\gamma=0.01$: 21.21
>     - $\gamma=0.1$: 18.72
>     - $\gamma=1$: 16.96
>     - training speed: 17.45 seconds / 1K images
>
> In this way, for Cats $256^2$, higher patch resolution steadily gives better results if one has the resource capacity to tune hyper-parameters for it (which was not the case for us). However, it comes with a considerably higher training cost.
> Please note, that StyleGAN2-ADA/StyleGAN3/EG3D also do not have robustness with respect to R1 penalty weight $\gamma$: e.g., for StyleGAN2-ADA (the most robust one) we obtained FID of 4.5 vs FID of 3.8 on FFHQ $256^2$ for $\gamma = 0.1$ and $\gamma = 0.05$.
> Our preliminary experiments showed that architectural variations are comparable for a fixed $\gamma$ when the dataset and the resolution are also fixed.
>
> =====================================================================
>
> **Q**. *It is good to see that the proposed generators can generate up to 512x512 images. I wonder what's maximum is. Can it be scaled to 1024x1024?*
>
> **A**.
> In our project, we launched just a single experiment somewhere close to the submission deadline to train our generator on FFHQ $1024^2$ with $128^2$ patch resolution. It gave us an FID of 20.33 at 3.5M processed images ($\approx$ 15\% of overall training time) which is good performance, but then the image quality had not been improving till 15M processed images. We decided to stop the training to free the resources for more urgent experiments since we hypothesized that we would need to increase the tri-plane resolution for it and likely tune some hyper-parameters.
> We believe that patch-wise training is applicable at high resolution ([AnyResGAN](https://chail.github.io/anyres-gan/) gives some ground to this claim), and we plan to explore this in the future.
>
> **Q**. *For face and cat, the camera poses are limited to the front views. Can you do large view changes, e.g., rendering side faces or rendering from overhead? I want to know whether the radiance field will remain view-consistent for unseen training views.*
>
> **A**.
> Since we do not condition the color prediction branch on view angles (similar to $\pi$-GAN, EG3D, and other prior works), it always remains view-consistent, even at extreme view angles (up to the randomness in the integral calculation in volume rendering).
> In section 3 of [this web page](https://rethinking-3d-gans.github.io/additional-results), we provide the visualizations for our method and also for EG3D on FFHQ for the camera positions 3 standard deviations away from the frontal pose: i.e., $\pm 3 \sigma$ right/left and $\pm 3 \sigma$ up/bottom which is $\pm$0.9 and $\pm$0.6 radians, respectively.
> For EG3D, we generated the samples using the official checkpoint and sampling scripts.
> Note that EG3D uses a re-aligned/re-cropped version of FFHQ and renders from a closer distance, which hides potential artifacts in the back of the head.

---

> > ### Comment · Reviewer_LXyr · 2022-08-09
> > **Thank you for the detailed rebuttal.**
> >
> > Thanks for the reply. I would encourage the authors to add discussion on sampling strategy in the paper.  The rebuttal answers my questions and I would like to keep the original rating.

---

> ### Author Response · Authors · 2022-08-05
> **Official authors' response [Part 1/2]**
>
> We appreciate the effort the reviewer spent studying our work and their positive assessment of our contributions.
> Below, we did our best to elaborate on all the raised questions, provide additional visualizations and perform additional experiments to address all the concerns.
>
> **Q**. *As shown in the appendix, the patch-wise training strategy is subpar compared to full-resolution training for 2D generation, which limits its adoption.*
>
> **A**.
> % While it's true that our results for patch-wise training of StyleGAN2 were subpar compared to full-resolution training, one needs to note that
> Patch-wise optimization of 2D/3D generators is currently in its infancy, and there are \textbf{very} few works which explore it.
> In our project, we investigated two of its important aspects: scale sampling strategies and the adaptation of discriminator's filters' to different image scales and developed two ideas to improve them.
> While it's true that our two ideas are not enough to close the gap between patch-wise training and full-resolution training, they make solid steps into bridging it.
> We believe that patch-wise optimization is promising and will be explored further, especially in the context of 3D and video generation (which is traditionally very expensive), and our ideas can find their use there.
> One important aspect which could be improved (and which we couldn't make work in our project) is providing the global image information to the discriminator when it processes small-scale patches: right now, it is forced to judge only on the local information, which turn out to be a huge limitation compared to a full-resolution discriminator.
>
> Also, note that [AnyResGAN](https://chail.github.io/anyres-gan/), a contemporary work, also uses patch-wise optimization, but for 2D generation and with a much larger patch resolution (256$^2$).
> And their patch-wise StyleGAN3 is very close to the full-resolution StyleGAN3 in terms of performance: FID of 3.95 vs FID of 3.06 on FFHQ $1024^2$ (Appx 7.5).
> They explored two completely different aspects of it: knowledge distillation from a coarse-scale teacher and generator conditioning on patch parameters.
> Hence, we believe patch-wise training is promising and its gap with full-resolution training will get smaller and smaller with time.
>
> =====================================================================
>
> **Q**. *Were there any strategies used to mitigate aliasing in generated patches (e.g., Mip-NeRF-like volume rendering)?*
>
> **A**.
> Well, that is an excellent question, and we must admit that we did not take care of aliasing effects.
> A straightforward (and "ideal") solution would be to generate a full-resolution image and then extract the patch with anti-aliasing, but it would be too expensive.
> Integrating the ideas from Mip-NeRF would: 1) introduce additional complexities into the project; and 2) it is not clear how to use it on top of tri-planes, since the tri-plane representation uses coordinates as the interpolation weights for plane features rather than the positional embeddings inputted to an MLP.
> This is why we didn't investigate anti-aliasing in our project and leave it as an important future research direction.
>
> =====================================================================
>
> **Q**. *How do you sample real patches? Were there any strategies used to mitigate aliasing?*
>
> **A**.
> We extract real patches from images using grid sampling with bilinear interpolation.
> Since we do not employ anti-aliasing techniques in the generator, we considered it to be proper not to use them for real patches either, since it would lead to a potential mismatch in real and fake distributions.
> Nevertheless, we agree that it is vital to explore it in the future, and it could significantly improve the performance of patch-wise training.

---

### Official Review · Reviewer_P8PQ · 2022-07-11

**Rating:** 5
**Confidence:** 3
**Soundness:** 3 good
**Presentation:** 2 fair
**Contribution:** 3 good

**Summary:**

The paper proposes a non-upsampler-based 3D-aware generator. To train the non-upsampler-based generator, the paper presents a patch-based discriminator. In contrast to previous patch-based schemes, the author improves patch-based discriminators in two ways. First, they adopt location and scale to modulate the discriminator (in particular, the feature maps).  Second, they modify the patch sampling strategy based on an annealed beta distribution to stabilize training and accelerate the convergence. Furthermore, they introduce two plants and food datasets, in order to evaluate 3D-aware generators.

**Questions:**

Please see the weaknesses.

**Limitations:**

Yes, they did.

**Strengths And Weaknesses:**

Strengths:

The paper is well-written, and the experiments are comprehensive.

The patch-based discriminator studied in the paper is important for differentiating high-resolution images and would be a good contribution to the community.

Weaknesses：

1. The paper's main contribution is to propose a new patch-based discriminator, which is particularly useful when the generator can only render partial pixels. I would like to know whether the generator synthesizes all images or only a subset of pixels during training. Since the generator adopts a tri-plane representation, it is memory efficient. If the generator is already able to synthesize all pixels without running out of memory, it seems unnecessary to adopt a patch-based discriminator.

2. I think patch-based discriminators are valuable. A more challenging setting can demonstrate the method's effectiveness, namely training pi-GAN on high-resolution images. Since pi-GAN is memory-intensive, rendering all pixels for high-resolution images is impossible during training. Maybe this setting is more suitable for demonstrating the efficacy of the patch-based discriminator.

3. If possible, the authors should provide the results of the full-resolution discriminator. Since the full-resolution discriminator can see the global image, its performance can be used as an upper bound for the reference of patch-based discriminators.

4. In equation 4, a simple baseline is $lerp[1, r/R, 1 - min(t/T, 1)]$. If possible, the author should compare this setting to the proposed beta strategy.

5. Line 204, how to use pose supervision for the discriminator? The author can briefly introduce the details.

---

> ### Author Response · Authors · 2022-08-05
> **Official authors' response [Part 2/2]**
>
> **Q**. *If possible, the authors should provide the results of the full-resolution discriminator. Since the full-resolution discriminator can see the global image, its performance can be used as an upper bound for the reference of patch-based discriminators.*
>
> **A**.
> A full-resolution discriminator would indeed be a reasonable upper bound for the performance.
> However, as described in the previous answers, it would be infeasible to train it for a 3D generator.
> In our project, while exploring different patch-wise training setups, we instead conducted many experiments on 2D generation on top of StyleGAN2-ADA.
> We reported the key findings in Appx A.1 — together with this full-resolution discriminator upper bound.
> What we found is that our current patch-wise training strategy still has a lot of room for improvement compared to the full-resolution training: patch-based StyleGAN2 (with the $64^2$ patch resolution) attains FID of 7.11 on FFHQ $512^2$ after 25M seen images versus FID of 3.83 for the full-resolution StyleGAN2.
> We believe that there are two reasons why it under-performs.
> First, it received less overall training signal if measured in the number of seen content, e.g., it has seen fewer "eyes variations", fewer "hair samples", etc. — and training it for longer indeed improves the performance: it attains FID of 4.76 after 100M seen images ($\times 4$ longer training);
> Second, in our current setup, we do not provide the information on global content to the discriminator when it judges small-scale patches.
> Several of our attempts to provide it did not work (see the Failed experiments section in Appx C), but we believe it to be a promising future research direction.
> Also, note that a concurrent patch-based [AnyResGAN](https://chail.github.io/anyres-gan/) comes very close to the performance of full-resolution generators — though in their case, the patch size is $256^2$.
>
> =====================================================================
>
> **Q**. *In equation 4, a simple baseline is $\text{lerp}[1, r/R, 1 - \min(t/T, 1)]$. The author should compare this setting to the proposed beta strategy if possible.*
>
> **A**.
> This is a good suggestion and we have launched 3 experiments on Cats $256^2$ for $T = 1000, 5000, 10000$.
> We report the results for it in Section 4 on [this web page](https://rethinking-3d-gans.github.io/additional-results).
> We called this sampling strategy "reversed uniform sampling": it first focuses on the full range of patch scales and then gradually decreases this range to coarse scales only.
> As one would expect, all the sampling strategies initially perform similarly during the initial training stage. However, then the reversed scale schedule starts to degrade in terms of performance because the discriminator starts operating on more coarse scales, making the generator forget high-frequency details, affecting FID.
> We thank the reviewer for this suggestion and include this exploration in the supplementary material.
>
> =====================================================================
>
> **Q**. *Line 204, how to use pose supervision for the discriminator?*
>
> **A**.
> For pose supervision, we take the rotation and elevation angles, encode them with positional embeddings and feed them into a 2-layer MLP.
> After that, we multiply the obtained vector with the last hidden representation in the discriminator, following the default Projection GAN strategy from StyleGAN2-ADA.
> This is a default strategy from EG3D but for their model, the authors condition on $4\times4$ camera extrinsic + $3\times 3$ camera intrinsics parameter matrices, obtained from an off-the-shelf estimator.
> In our case, it is just 2 scalars.
> We added these details into the main text.
> But also note that we release the source code, where any further technical details could also be found.

---

> ### Author Response · Authors · 2022-08-05
> **Official authors' response [Part 1/2]**
>
> We are thankful for the review and the valuable suggestions, and we agree that our developed patch-based discriminator would be a good contribution to the community.
> In the following, we elaborate on the raised questions, clarify some misunderstandings and provide the results for the additional experiments.
>
> =====================================================================
>
> **Q**. *Does the generator synthesize all images or only a subset of pixels during training? If the tri-plane generator is already able to synthesize all pixels without running out of memory, it seems unnecessary to adopt a patch-based discriminator.*
>
> **A**.
> Our generator synthesizes just $64^2$ pixels for each random sample in a batch during training instead of $512^2$ (or $256^2$) ones, which leads to a greatly improved training speed.
>
> Synthesizing $512^2$ pixels on each iteration is not computationally feasible because tri-planes are still very expensive in high resolutions.
> EG3D trains with $64^2$ resolution tri-planes for $25M$ images and then increases this resolution to $128^2$ and fine-tunes for $1.5M$ images.
> This slight increase in the resolution significantly decreases the training speed: from 24 seconds per 1K images to 46 seconds per 1K images (as per Appx 3 in [EG3D](https://nvlabs.github.io/eg3d/media/eg3d.pdf)) on 8 $\times$ v100 GPUs.
> So, generating these additional $128^2 - 64^2 = 12288$ pixels in NVidia's tri-plane implementation cost ${\approx}22$ seconds per 1K images.
> In this way, training a full-resolution tri-plane-based generator at $512^2$ image size would take **493.3 seconds per 1K images** (we consider the cost of the 2D upsampler to be negligible here).
> The overall training time till 25M images are processed (the standard schedule for StyleGAN-1/2/3, EG3D, StyleNeRF, our generator, and other models) would thus cost **${\approx}$4.5 months** on $8\times$ NVidia V100 GPUs, which is far beyond the resource capacity of most research teams.
>
> =====================================================================
>
> **Q**. *Maybe implementing patch-wise training on top of $\pi$-GAN is more suitable for demonstrating the efficacy of the patch-based discriminator.*
>
> **A**.
> As said in the previous answer, tri-planes are still a good test-bed for exploring patch-wise training since they are also very expensive to scale to high resolutions. But, indeed, exploring patch-wise training on top of MLP-based NeRF generators (like $\pi$-GAN) is an exciting direction.
>
> We implemented a $\pi$-GAN generator in our repo with patch-wise training for this discussion.
> As the generator, we used an 8-layer MLP with 256 channels and positional embeddings of the coordinates (the setup from the original NeRF paper).
> We had to decrease the patch size to $32^2$ from $64^2$ to make it train faster due to the time limit (7.2 seconds per 1K images instead of 18.3).
> Furthermore, we also used 24 steps per ray in volumetric rendering.
> The attained FID after 10M seen images (1 day of training on 8$\times$ v100s) is 21.46: for comparison, the original $\pi$-GAN obtains FID of 68.28 after full training, which takes 1 week on $8\times$ V100 at the $256^2$ resolution.
> We provide the samples for this model in Section 6 of the [this web page](https://rethinking-3d-gans.github.io/additional-results).
> Note that FID is greatly affected by the $\pi$-GAN's circular artifacts in the generations (see the [original samples](https://marcoamonteiro.github.io/pi-GAN-website/) on Cats).

---

> ### Author Response · Authors · 2022-08-08
> **A question on whether any other potential concerns are left**
>
> Dear Reviewer, we are very thankful for your feedback which helped us to improve several important parts of our work. And if that's possible, it would be crucial for us to know if there are any concerns left on your side after our response? As far as we understood, the main concern of the review originates from the surmise that a tri-plane generator could be trained in full-resolution on its own, which eliminates the need for patch-wise training. In our response, we provided an exposition showing that training it in full resolution would result in ~$20\times$ longer training, making it computationally infeasible *without* our patch-wise scheme.
>
> We also conducted an additional series of experiments to ablate the proposed scale sampling strategy; implemented and conducted an experiment with the pi-GAN-based generator; and elaborated on pose conditioning in the discriminator. And we would be happy to know whether our exposition is convincing and whether there is anything else we could elaborate on to resolve the existing or any new concerns. And once again, we apologize for submitting our response with the delay.

---

### Official Review · Reviewer_vtTi · 2022-07-11

**Rating:** 7
**Confidence:** 3
**Soundness:** 3 good
**Presentation:** 3 good
**Contribution:** 3 good

**Summary:**

This paper introduces new techniques in training NeRF-based GANs. The proposed techniques addresses the patch-based trianing issue, which blocked privious works to directly generate nerfs that renders into high quality resolution images. As a result, previous works lacks in training time, in addition to lack of view consistency because of image-based upsampling. The proposed method utilizes a hyper-network to generate filters for the patch discriminator under different resolutions, and utilizes an annealed beta distribution for samping the random scale for patch discrimination. Both disign choices are well founded and validated by the quality of the result.

**Questions:**

1. Are all baseline results generated using view-independent NeRF as well?
2. As suggested by the paper, using the beta distribution with the improved annealing improves the stability and convergence speed. Is that also the case for the hyper-network as well? Is the method robust to train given the hyper-generated patch discriminator?




**Limitations:**

The authors addressed the limitations and potential negative impact adequately.

**Strengths And Weaknesses:**

Strengths:
Originality:
This paper proposes new solutions to patch-based discriminators in training NeRF-based GANs. The proposed techniques are simple to implement and effective, as demonstrated by the results. In addition, the authors validate the quality on new datasets(megascan plants&food), which further demonstrated the geometric quality of the generated nerf.

Significance:
I think this paper would be of significance to communities interested in generative methods centered around NeRF. Though it can be said that the proposed technique is simple, but it is more a merit than a drawback. I think those solid gadgets are keys to make things work better and better.

Quality:
This paper conducts enough experiments to support the claim, and justifications of the design choices make sense. I find the evidence in the paper convincing and the webpage of results are quite representative.

Clarity:
This paper is well written and easy to follow. Readers should be able to reimplement the method based on the inforamtion provided.

---

> ### Author Response · Authors · 2022-08-05
> **Official authors' response**
>
> We are delighted to receive such a positive assessment of our work and are grateful for it.
> Below, we provide the answers to the raised questions with a reference for additional results.
>
> =====================================================================
>
> **Q**. *Are all baseline results generated using view-independent NeRF as well?*
>
> **A**.
> That's true, **all** the generators are trained without the view dependence.
> In our preliminary experiments, we observed that since there is not enough supervision in terms of views (e.g., FFHQ and Cats have just a single view per object) and there exist 3D biases in the existing benchmarks (e.g., frontal views in FFHQ have much more smiling people than side views) — the multi-view consistency severely degrades.
> Prior works also disable the view dependence (e.g., $\pi$-GAN, EG3D, GRAM, and others.)
>
> =====================================================================
>
> **Q**. *Is the method robust to train with the proposed beta sampling strategy given the hyper-generated patch discriminator?*
>
> **A**.
> We were unclear in our exposition: Figure 6 shows the improved convergence for the hyper-conditioned rather than standard discriminator — i.e., our patch sampling scheme does improve optimization in such a setup.
> To address the inaccuracy, we performed the same ablation on Cats $256^2$ for the standard discriminator and reported them in Section 5 on [this web page](https://rethinking-3d-gans.github.io/additional-results).
> We also observed the same effect in this scenario: the beta scale sampling strategy made the training more robust and convergence faster.

---

> > ### Comment · Reviewer_vtTi · 2022-08-08
> > **Thanks for the response**
> >
> > Thank you for the detailed response to all the questions. I find them quite informative and would maintain my original rating.

---

### Author Response · Authors · 2022-08-02
**Delay in authors' response**

We are very grateful to the reviewers for their careful analysis of our work and all the suggestions they gave. While preparing our response, we ran into an embarrassing misfortune: our internal cluster got shut down for maintenance for 5 days, and it took us a huge effort to migrate to a new external cluster. We are currently running the necessary experiments and preparing the visualizations/comparisons to address all the raised questions and concerns — and will post them with some short delay. We hope that this delay will not be too much of an inconvenience to the reviewers.

---

### Author Response · Authors · 2022-08-05
**Update summary**

We thank the reviewers for their feedback — it helps us to improve the work and gives good ideas on what directions to explore in the future.
We sincerely apologize for the delay with our response and now provide the additional experiments, comparisons, and clarifications for our model.
We set up a separate anonymous web page to host the necessary media files for this: [https://rethinking-3d-gans.github.io/additional-results](https://rethinking-3d-gans.github.io/additional-results) (we will also include them in the supplementary material/web page).
Here is the summary of what we did:
- Elaborated on all the raised questions and concerns.
- Provided geometry and multi-view consistency comparisons on FFHQ $512^2$ with EG3D, as suggested by R4, and extreme-angles visualizations on FFHQ $512^2$, as suggested by R3.
- Conducted 9 additional experiments on Cats $256^2$ for different patch sizes and R1 $\gamma$ regularization weights for a more thorough analysis of the patch resolution influence.
- Conducted 3 additional experiments on Cats $256^2$ with the $\text{lerp}[1, r/R, 1 - min(t/T, 1)]$ patch scale sampling strategy, as suggested by R2.
- Conducted 4 additional experiments on Cats $256^2$ for the standard discriminator for different scale sampling strategies to address the concerns of R1 and R4.
- Conducted 2 additional experiments for the bare tri-planes + GRAF uniform scale sampling setup on FFHQ $512^2$ and M-Plants $256^2$, as suggested by R4.
- Launched EG3D on the standard FFHQ $512^2$ but observed that it seems to require some additional tuning to obtain the same performance.
- Implemented a simple MLP-based NeRF generator with patch-wise training in our code repo and tested it on Cats $256^2$: it attained FID of 21.46 after 1 day of training on 8 $\times$ V100 GPUs (compared to FID of 68.28 for $\pi$-GAN after 7 days of training).
- We invested additional efforts into tuning the hyperparameters for GRAM and managed to obtain a ${\approx}$20\% better performance for it on M-Plants $256^2$ and M-Food $256^2$. However, it was still diverging into a mode collapse.
- Trained StyleNeRF with adaptive differentiable augmentation on Cats $256^2$ and observed that it dramatically improved its performance: from FID of 27.91 to FID of 5.91.
- We made some minor changes to the submission text.

We updated our submission to reflect the changes (highlighted in blue) and will include all the additional results in the supplementary and/or on the accompanying web page.
Once again, we apologize for the delayed response and hope to have a fruitful discussion.

---

### Meta-Review · Area_Chair_Wi3M · 2022-08-26

**Recommendation:** Accept
**Confidence:** Certain

**Metareview:**

The reviewers found the method simple and effective and considered it a contribution of interest to the community. Claims are well supported by experiments and design choices have been validated. The paper is well written. Furthermore, the authors provided highly detailed responses to all questions by reviewers, which creates confidence that reviewers' remarks will be addressed in the final paper.

**Award:**

No

---

### Decision · Program_Chairs · 2022-09-14

Accept